# Server-Side Stepsizes and Sampling Without Replacement Provably Help in Federated Optimization

## Abstract

We present a theoretical study of server-side optimization in federated learning. Our results are the first to show that the widely popular heuristic of scaling the client updates with an extra parameter is very useful in the context of Federated Averaging (FedAvg) with local passes over the client data. Each local pass is performed without replacement using Random Reshuffling, which is a key reason we can show improved complexities. In particular, we prove that whenever the local stepsizes are small, and the update direction is given by FedAvg in conjunction with Random Reshuffling over all clients, one can take a big leap in the obtained direction and improve rates for convex, strongly convex, and non-convex objectives. In particular, in non-convex regime we get an enhancement of the rate of convergence from $\mathcal{O}\left(\varepsilon^{-3}\right)$ to $\mathcal{O}\left(\varepsilon^{-2}\right)$. This result is new even for Random Reshuffling performed on a single node. In contrast, if the local stepsizes are large, we prove that the noise of client sampling can be controlled by using a small server-side stepsize. To the best of our knowledge, this is the first time that local steps provably help to overcome the communication bottleneck. Together, our results on the advantage of large and small server-side stepsizes give a formal justification for the practice of adaptive server-side optimization in federated learning. Moreover, we consider a variant of our algorithm that supports partial client participation, which makes the method more practical.

## 1 Introduction

The unprecedented industrial success of modern machine learning techniques, tools and models can to a large degree be attributed to the abundance of data available for training. Indeed, the most popular and best performing deep learning models rely on a very large number of parameters, and in order to generalize well, need to be trained using optimization algorithms over very large training datasets. Other things equal, the more data we have, the better. A key driving force behind the proliferation of such data is the massive digitization of society of the last few decades. People have access to increasingly more elaborate personal and home smart devices capable of generating, capturing and processing data such as text, images and videos. Similarly, in the sphere of governments and corporations, much of what used to be done through a physical exchange (e.g., via paper/fax/letter) is now performed in a digital form, generating treasure troves of potentially useful data. For example, hospitals collect, store and make us of a variety of patient data, ranging from routine bodily functions to PET scans and genome sequencing.

### 1.1 Federated learning

The traditional way of learning from this data is to collect it in a single (and often proprietary) data center, where it is subsequently processed using modern machine learning algorithms. However, due to several considerations which keep gaining in importance, such as energy efficiency and privacy, it is often desirable to avoid centralized training altogether, and instead perform the training without the data ever leaving the clients' secure sites. Introduced in 2016 by Konečný et al. (2016); Konečný et al. (2016); McMahan et al. (2017), this is precisely the promise and subject of study of *federated learning (FL)*. In other words, federated learning means efficient machine learning over data stored in a distributed fashion across a network

of heterogeneous clients (e.g., mobile phones, smart devices, companies) that captured and own the data, using these clients' machines/devices not only as data sources, but also as computers that contribute to the training.

## 1.2 Problem formulation

We consider the standard optimization formulation of federated learning

$$\min_{x \in \mathbb{R}^d} \left[ f(x) \stackrel{\text{def}}{=} \frac{1}{M} \sum_{m=1}^{M} f_m(x) \right], \tag{1}$$

where $M$ is the total number of clients, $x \in \mathbb{R}^d$ represents the parameters of the model we wish to train, and $f_m : \mathbb{R}^d \to \mathbb{R}$ is the loss of model $x$ on the training data owned by client $m \in [M] \stackrel{\text{def}}{=} \{1, 2, \dots, M\}$. Typically, $M$ is very large.

Since the training dataset on each client is necessarily finite, we assume that $f_m$ has the finite-sum structure

$$f_m(x) \stackrel{\text{def}}{=} \frac{1}{n} \sum_{i=1}^{n} f_m^i(x), \tag{2}$$

where $f_m^i : \mathbb{R}^d \to \mathbb{R}$ is the loss of model $x$ on training example $i \in [n] \stackrel{\text{def}}{=} \{1, 2, \dots, n\}$ stored on client $m$. We assume that the functions $f_m^i$ are differentiable, and consider the strongly convex, convex and non-convex regimes.

## 1.3 Ingredients of successful federated learning methods

Practical considerations of federated learning systems and vast experimental evidence accrued over the last few years point to several design constraints and algorithmic ingredients which have proved useful in the context of federated learning methods for solving equation 1-equation 2. We now very briefly outline some of them. More details can be found in the appendix where we review related work.

**Partial participation.** In federated learning, training is performed through several communication rounds in each of which an orchestrating server chooses a *cohort* of clients that will be participating in the training process in that round. This practice is known as *partial participation*, and is necessary due to practical considerations and limitations, such as limited server capacity, and limited client availability (Kairouz et al., 2021). However, partial participation can be useful also due to the diminishing returns one gets as the number of participating clients grows (Charles et al., 2021). Partial participation is a necessity in the cross-device regime where the training is performed over a very large number of clients (i.e., $M$ is very large) most of which will only participate in the entire training procedure at most once. Sampling of clients to form a cohort can be done adaptively so as to choose the most informative clients (Chen et al., 2020).

**Local training.** At the beginning of each communication round, each client in the cohort is provided with the latest model by the orchestrating server, which is used as a starting point for *local training*. Local training refers to the common practice in FL of performing several steps of a suitably chosen local optimization procedure, such as one of the many variants of SGD, using its own local training data. Perhaps the simplest approach is to perform a single local GD iteration. If the model updates are simply just aggregated by the server, then the resulting method can be seen as Minibatch SGD, where the minibatches correspond to the cohorts. However, it is typically more efficient to perform *multiple* local steps (McMahan et al., 2017), and to use local optimizers that rely on *incremental* data processing, such as SGD.

**Data shuffling.** Typically, the local training dataset is processed once or several times in an incremental fashion; that is, one data point (or one small minibatch) at a time. However, experimental evidence shows that processing the local data *without replacement* can lead to substantially better results than processing the data *with replacement*. In particular, processing the local training data in an order dictated by a random permutation—a technique known as Random Reshuffling (RR)—is often set as default in modern deep learning and federated learning software (Bottou, 2009; Bengio, 2012; Sun, 2020). This is in sharp contrast with the *with-replacement* sampling of data employed by SGD. With-replacement sampling ensures that the gradient

Table 1: Conceptual comparison of results for FedAvg from prior work with our results.

| Partial participation | Local training | Data shuffling | **Large** server stepsizes help | **Small** server stepsizes help | Reference |
|:---:|:---:|:---:|:---:|:---:|:---|
| ✓ | ✓ | ✗ | ✓ | ✗ | Karimireddy et al. (2020) |
| ✓ | ✓ | ✗ | ✓ | ✗ | Woodworth et al. (2020) |
| ✗ | ✓ | ✗ | ✗ | ✗ | Koloskova et al. (2020) |
| ✗ | ✓ | ✗ | ✗ | ✗ | Khaled et al. (2020) |
| ✗ | ✓ | ✓ | ✗ | ✗ | Mishchenko et al. (2021) |
| ✓ | ✓ | ✓ | ✓ | ✓ | **This paper** |

updates are unbiased, and this simplified the analysis. For this reason, SGD is significantly better understood in theory than its better performing but much more poorly understood cousin RR. However, recent results of Mishchenko et al. (2020), and extensions due to Mishchenko et al. (2021) and Yun et al. (2021) to distributed training, show that RR can have clear theoretical advantages over SGD.

**Server stepsizes.** Once local training is finished, the clients in the cohort send their models or model updates to the orchestrating server, which typically aggregates them via averaging. This information is then used to perform *server side* optimization. The simplest approach is to do nothing; that is, to treat the aggregated models as the next global model that is broadcast to the new cohort in the next communication round. However, empirical evidence suggests that it is better to aggregate *model updates*, and treat them as gradient-type information which can be injected into a suitably chosen server side optimization routine (Karimireddy et al., 2020). For example, the server may run one step of GD using the aggregated model update as a proxy for the gradient which is not available, with its own server-side stepsize.

**Further useful tricks.** Additional tricks that are often employed in the context of federated learning include the use of compressed communication (Alistarh et al., 2018; Gorbunov et al., 2021), drift reduction (Karimireddy et al., 2020; Gorbunov et al., 2020), error compensation (Stich & Karimireddy, 2019; Richtárik et al., 2021), server side momentum (Hsu et al., 2019), and adaptive stepsize selection (Reddi et al., 2020). These techniques are beyond the scope of this paper.

## 2  Summary of Contributions

Despite the fact that *partial participation*, *local training*, *data shuffling* and *server stepsizes* have all been empirically found to be very useful building blocks of FL methods, most of these techniques are not very well understood in theory even in isolation. Informally speaking, and at the risk of oversimplifying the current state of affairs, we know virtually nothing about *server stepsizes*, very little about *data shuffling*, relatively much more about *local training*, and quite a bit, but still "not enough", about *partial participation*.

> The key focus of this paper is to make a substantial advance in the current theoretical understanding of *server stepsizes* in the context of *realistic* federated learning.

In order to theoretically understand the server stepsize phenomenon in a realistic context of techniques commonly used in FL, we study this phenomenon *together* with data shuffling, local training and partial participation. While this makes the analysis substantially harder and different from all[1] existing analyses of FedAvg, we believe it is important to do so as this will highlight the *interplay* between these algorithmic techniques and their *combined* impact on training.

A brief visual summary of this in the context of selected existing methods is provided in Table 2. We summarize our contributions as follows:

• **New algorithm.** We design a new algorithm, for which we coin the name Nastya (Algorithm 1; see Section 4), which combines all the of the aforementioned practical tricks and techniques in a single method:

---
[1]Except for the recent work of Mishchenko et al. (2021) which we used as an inspiration.

Table 2: Comparison of convergence results for FedAvg from prior work with our results.

| Method | Strongly convex[2] | Non-convex | Reference |
|---|---|---|---|
| SCAFFOLD [1] | $\tilde{\mathcal{O}}\left(\frac{\sigma^2}{\mu M n \epsilon} + \frac{1}{\mu}\right)$ | $\mathcal{O}\left(\frac{\sigma^2}{M n \epsilon^2} + \frac{1}{\epsilon}\right)$ | Karimireddy et al. (2020) |
| Local SGD [1] | $\tilde{\mathcal{O}}\left(\frac{L}{\mu} + \frac{\sigma^2}{M\mu\varepsilon} + \sqrt{\frac{Ln(\sigma^2+n\zeta^2)}{\mu^2\varepsilon}}\right)$ [3] | ✗ | Woodworth et al. (2020) |
| Local SGD | $\tilde{\mathcal{O}}\left(\frac{\sigma_*^2}{M\mu\epsilon} + \frac{\sqrt{L}(n\zeta+\sqrt{n}\sigma)}{\mu\sqrt{\epsilon}} + \kappa n\right)$ [3] | $\mathcal{O}\left(\frac{L\sigma_*^2}{M\epsilon^2} + \frac{L(n\zeta+\sqrt{n}\sigma)}{\epsilon^{3/2}} + \frac{Ln}{\epsilon}\right)$ [3] | Koloskova et al. (2020) |
| FedRR | $\tilde{\mathcal{O}}\left(\frac{L}{\mu} + \frac{\sqrt{\kappa n}(\sigma_*+\sqrt{n}\zeta)}{\mu\sqrt{\varepsilon}}\right)$ [3] | ✗ | Mishchenko et al. (2021) |
| Nastya | $\tilde{\mathcal{O}}\left(\frac{Ln}{\mu}\right)$ | $\mathcal{O}\left(\frac{Ln}{\varepsilon}\right)$ | **This paper** |

[1] The analysis is done under the bounded variance assumption: $g_i(x) := \nabla f_i(x;\zeta_i)$ is unbiased stochastic gradient of $f_i$ with bounded variance $\mathbb{E}_{\zeta_i}\left[\|g_i(x) - \nabla f_i(x)\|^2\right] \leq \sigma^2$, for any $i, x$.

[2] The $\tilde{O}$ notation omits $\log\frac{1}{\varepsilon}$ factors

[3] Here we use $\zeta^2 \overset{\text{def}}{=} \frac{1}{M}\sum_{m=1}^{M}\|\nabla f_m(x_*)\|^2$.

*partial participation*, *local training*, *data shuffling* and, most importantly, *server stepsizes*. In our method, in each communication round $t$, the cohort is chosen as a random subset $S_t$ of the set $\{1, 2, \ldots, M\}$ of clients of cardinality $1 \leq C \leq M$, chosen uniformly from all subsets of cardinality $C$. Each device performs local training via a single pass of incremental GD with *client stepsize $\gamma > 0$* over the local training data points in an order dictated by a *random permutation*. We allow for two options: i) either the random permutation for all clients is sampled just once and used in all communication rounds (*Shuffle-Once* option), or ii) the random permutation is sampled afresh at the start of each communication round (*Random-Reshuffling* option). At the end of local training, the updated models are communicated back to the server, which uses these updates to form a *gradient estimator*, and applies one step of GD using a server stepsize $\eta > 0$ with this estimator in lieu of the true gradient. The new model is then broadcast to a new cohort in the next communication round, and the process is repeated.

• **Complexity analysis.** We provide strong complexity analysis of our new algorithm for strongly convex (Theorem 1), convex (Theorem 2) and non-convex (Theorem 3) functions; see Table **??**. This is the first theory for a variant of FedAvg that combines the benefits of partial participation, data shuffling, local training and, most importantly, also *server stepsizes*. Most importantly, with a couple exceptions only (Karimireddy et al., 2020; Woodworth et al., 2020), there are no prior theoretical works analyzing the effect of server stepsizes in FL. The methods in the aforementioned works use local training and partial participation, but do not use data shuffling, and are significantly different from ours.

• **Small client stepsizes, large server stepsizes, and no need for drift reduction.** In particular, Theorems 1, 2 and 3, covering the strongly convex, convex and non-convex regimes, respectively, suggest that the server can use the *large $\mathcal{O}(1/L)$* stepsize, where $L$ is the Lipschitz constant of the gradient of $f$. In the strongly convex and convex regimes, based on our theory, it is optimal for the client stepsize $\gamma$ to be *small*, which completely eliminates the second of the three terms in the complexity bounds (see the third column of Table **??**) which controls the price one pays due to *data heterogeneity*. Indeed, our theory allows for the client stepsize $\gamma$ to be small while the server stepsize $\eta$ can be large (see the second column of Table **??**).

Note that in all three regimes, and thanks to the fact that we employ a *data shuffling* strategy, this second term depends on the square $\gamma^2$ of the client stepsize, which means that we can make this term small without making the client stepsizes infinitesimal. So, thanks to Nastya's use of data shuffling strategies, it does *not* require any explicit drift reduction technique such as SCAFFOLD to handle data heterogeneity (Karimireddy et al., 2020).

• **Small server stepsizes can be beneficial.** To the best of our knowledge, no prior theoretical work suggests that it might be beneficial to use *small* server stepsizes. Our results (see Theorem 5) suggest that

this can be the case when each $f_m^i$ is strongly convex and smooth, and when the strong convexity parameter is very small.

• **Experimental validation of our theoretical predictions.** We provide experimental examination of Nastya and compare it with selected benchmarks. Our goal is not to perform large scale experiments and claim empirical superiority because the algorithmic ingredients embedded in Nastya already *are* being used in practical FL methods precisely because they have already been empirically found to be useful. This allows us to focus on simple experiments which test the theoretical predictions of our theory.

Our experimental results confirm our theory, and illustrate the behavior of the methods we test in various settings. Moreover, we go beyond the theory and conduct additional experiments with the adaptive stepsize strategy introduced by Malitsky & Mishchenko (2020). Inspired by Reddi et al. (2020), we additionally utilize several server-side optimization subroutines on top of the local updates.

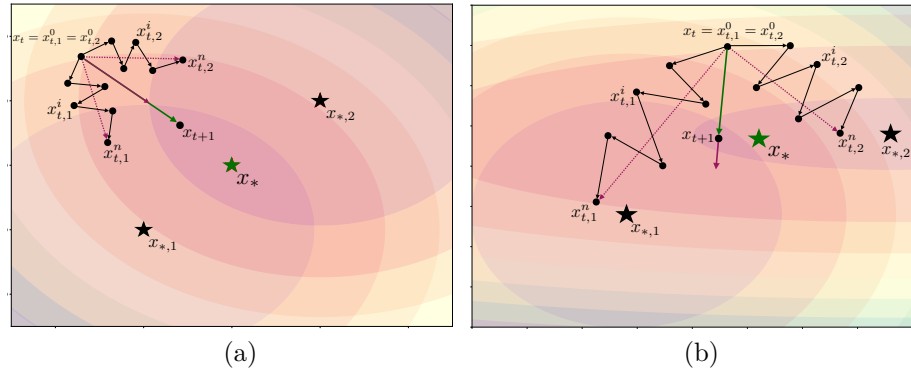

(a)            (b)

Figure 1: Illustration of the dependence between server and client stepsizes on a simple example with $M = 2$ clients. $x_{*,1}$ and $x_{*,2}$ are the minimizers of the local functions $f_1$ and $f_2$, respectively, and $x_*$ is the minimizer of the global function $f = \frac{1}{2}f_1 + \frac{1}{2}f_2$. **(a)** In the case of small client stepsizes $\gamma$, the average of local steps is not large, but at the same time the variance is small and the direction is close to direction of the full gradient, which allows us to go further towards this direction by employing a large server stepsize $\eta$. **(b)** In the case of large client stepsizes $\gamma$, each client step contributes to the global step, but the variance grows as well, so it is useful to use smaller server stepsize $\eta$ to reduce this variance. These intuitions are confirmed by our theory.

## 3 Preliminaries

In this section we introduce several key concepts that will help us to formulate our theoretical results. In all our theoretical results we rely on smoothness, and in some we require convexity or strong convexity.

**Definition 1** (*L*-smoothness). Function $\phi \colon \mathbb{R}^d \to \mathbb{R}$ is *L*-smooth if it has *L*-Lipschitz continuous gradient for some $L > 0$

$$\|\nabla\phi(x) - \nabla\phi(y)\| \le L\|x - y\| \quad \forall x, y \in \mathbb{R}^d. \tag{3}$$

**Definition 2** (Convexity and strong convexity). Function $\phi \colon \mathbb{R}^d \to \mathbb{R}$ is convex if $\forall x, y \in \mathbb{R}^d$

$$\phi(y) \ge \phi(x) + \langle \nabla\phi(x), y - x \rangle, \tag{4}$$

and $\mu$-strongly convex if $\forall x, y \in \mathbb{R}^d$

$$\phi(y) \ge \phi(x) + \langle \nabla\phi(x), y - x \rangle + \tfrac{\mu}{2}\|y - x\|^2. \tag{5}$$

In our analysis we use the following assumption.

**Assumption 1.** The objective $f$ and the individual losses $f_m^1, \ldots, f_m^n$ are all *L*-smooth. Further, for all $i$ and $m \in \{1, 2, \ldots, M\}$ and $i \in \{1, 2, \ldots, n\}$, (i) $f_* \stackrel{\text{def}}{=} \inf_x f(x) > -\infty$, (ii) $f_{*,m} \stackrel{\text{def}}{=} \inf_x f_m(x) > -\infty$,

and (iii) $f_{*,m}^i \overset{\text{def}}{=} \inf_x f_m^i(x) > -\infty$. If $f_m^i$ is convex, we further assume the existence of minimizers $x_* = \arg\min_{x \in \mathbb{R}^d} f(x)$ and $x_{*,m}^i = \arg\min_{x \in \mathbb{R}^d} f_m^i(x)$.

While our theory does not require any *assumptions* on data homogeneity, our *results* will reflect the degree to which the data are heterogeneous, and are better for data that are "more" homogeneous. In particular, in the strongly convex and convex regimes we rely on the following notions.

**Definition 3** (Variance at the optimum). The variance of the gradients $\{\nabla f_m\}_{m=1}^M$ at $x_*$ is defined as

$$\sigma_*^2 \overset{\text{def}}{=} \frac{1}{M} \sum_{m=1}^{M} \|\nabla f_m(x_*)\|^2,$$

where $x_*$ is a minimizer of $f$. The variance of the gradients $\{\nabla f_m^i\}_{i=1}^n$ at $x_*$ is

$$\sigma_{*,m}^2 \overset{\text{def}}{=} \frac{1}{n} \sum_{i=1}^{n} \|\nabla f_m^i(x_*)\|^2.$$

An important lemma that allows us to obtain a strong upper bound for variance in the case of sampling without replacement, which our data shuffling methods rely on, was formulated by Mishchenko et al. (2020). We include it here for completeness.

**Lemma 1** (Sampling without replacement). Let $X_1, \ldots, X_n \in \mathbb{R}^d$ be fixed vectors, $\overline{X} \overset{\text{def}}{=} \frac{1}{n} \sum_{i=1}^n X_i$ be their average and $\sigma^2 \overset{\text{def}}{=} \frac{1}{n} \sum_{i=1}^{n} \|X_i - \overline{X}\|^2$ be the population variance. Fix any $k \in \{1, \ldots, n\}$, let $X_{\pi_1}, \ldots X_{\pi_k}$ be sampled uniformly without replacement from $\{X_1, \ldots, X_n\}$ and $\overline{X}_\pi$ be their average. Then, it holds

$$\mathbb{E}\left[\overline{X}_\pi\right] = \overline{X}, \quad \mathbb{E}\left[\|\overline{X}_\pi - \overline{X}\|^2\right] = \frac{n-k}{k(n-1)}\sigma^2. \tag{6}$$

For non-convex functions, we use a different notion of data heterogeneity.

**Definition 4** (Functional dissimilarity). The variance at the optimum in the non-convex regime is defined as

$$\Delta_* \overset{\text{def}}{=} f_* - \frac{1}{M} \sum_{m=1}^{M} f_{*,m},$$

where $f_{*,m} = \inf_x f_m(x)$ and $f_* = \inf_x f(x)$. For each device $m$, the variance at the optimum is defined as

$$\Delta_{*,m} \overset{\text{def}}{=} f_* - \frac{1}{n} \sum_{i=1}^{n} f_{*,m}^i,$$

where $f_{*,m}^i = \inf_x f_m^i(x)$.

Again, the above is a definition and not an assumption. The concepts are well defined as long as Assumption 1 is satisfied.

## 4 The Nastya Algorithm

We now formally describe our Nastya algorithm (see Algorithm 1). Nastya combines several techniques that were empirically found to be useful in FL: *partial participation*, *local training*, *data shuffling* and *server stepsizes*.

In each communication round $t \geq 0$ of Nastya, the cohort $S_t$ is chosen as a random subset of the set $\{1, 2, \ldots, M\}$ of all clients. In particular, we choose a random subset of cardinality $C$ (the cohort size), where $1 \leq C \leq M$, uniformly at random. The server then sends the global model $x_t$ to all clients in the cohort. Setting $C = M$ models the full participation regime.

---

**Algorithm 1** Nastya: Federated optimization with server stepsize, random shuffling and partial participation

---

1: **Input:** client stepsize $\gamma > 0$; server stepsize $\eta \geq 0$; cohort size $C \in \{1, 2, \ldots, M\}$; initial iterate/model $x_0 \in \mathbb{R}^d$; number of communication rounds $T \geq 1$
2: **Shuffle-Once option:** For each client $m$, sample a permutation $\pi_m = (\pi_m^0, \pi_m^1, \ldots, \pi_m^{n-1})$ of $\{1, 2, \ldots, n\}$
3: **for** communication round $t = 0, 1, \ldots, T-1$ **do**
4:      Sample a cohort $S_t$ of $C$ clients (server chooses a random set $S_t \subseteq \{1, 2, \ldots, M\}$ of size $|S_t| = C$, uniformly at random)
5:      Send model $x_t$ to all participating clients $m \in S_t$            (server broadcasts $x_t$ to all clients $m \in S_t$ )
6:      **for** all clients $m \in S_t$, locally in parallel **do**
7:          $x_{t,m}^0 = x_t$            (client $m$ initializes local training using the latest global model $x_t$)
8:          **Random-Reshuffling option:** Sample a permutation $\pi_m = (\pi_m^0, \pi_m^1, \ldots, \pi_m^{n-1})$ of $\{1, 2, \ldots, n\}$
9:          **for** all local training data points $i = 0, 1, \ldots, n-1$ **do**
10:              $x_{t,m}^{i+1} = x_{t,m}^i - \gamma \nabla f_m^{\pi_m^i}(x_{t,m}^i)$ (client $m$ makes one pass over its local training data in the order dictated by $\pi_m$)
11:          $g_{t,m} = \frac{1}{\gamma n}(x_t - x_{t,m}^n)$            (client $m$ computes local update direction $g_{t,m}$)
12:      $g_t = \frac{1}{C} \sum_{m \in S_t} g_{t,m}$          (server aggregates the local update directions $g_{t,m}$ discovered by the cohort $S_t$ of clients)
13:      $x_{t+1} = x_t - \eta g_t$          (server updates the model using the aggregated direction $g_t$ and applying server stepsize $\eta$)

---

Each participating client $m \in S_t$ then performs local training using a single pass of incremental GD with *client stepsize* $\gamma > 0$ over the local training data points in an order dictated by a *random permutation*

$$\pi_m = (\pi_m^1, \pi_m^2, \ldots, \pi_m^n)$$

of the indices of the local training dataset $\{1, 2, \ldots, n\}$. In particular, the following update is iterated for $i = 0, \ldots, n-1$:

$$x_{t,m}^{i+1} = x_{t,m}^i - \gamma \nabla f_m^{\pi_m^i}(x_{t,m}^i),$$

where $x_{t,m}^0$ is initialized to $x_t$, and $\gamma > 0$ is the client stepsize. That is, we run one pass over the local data using the RR method (Mishchenko et al., 2020). This differs from one pass over the data via SGD in that each data point is sampled exactly once.

Note that we allow for two options for how the permutation is formed: i) either the random permutation is sampled just once for all clients, and used in all communication rounds (*Shuffle-Once* option), or ii) the random permutation is sampled afresh at the start of each communication round (*Random-Reshuffling* option). Both have the same theoretical properties in our analysis.

At the end of local training, the updated models $x_{t,m}^n$ are communicated back to the server, which uses these updates to form a *gradient-type estimator* $g_t$, and applies one step of GD using a server stepsize $\eta > 0$ with this estimator in lieu of the true gradient. Equivalently and this is how we decided to formally state the method, each client $m \in S_t$ sends the following scaled model difference to the server:

$$g_{t,m} = \frac{1}{\gamma n}(x_t - x_{t,m}^n),$$

where $x_{t,m}^n$ is the model found by the client after one pass over the data via RR. The server then aggregates these vectors from all clients in the cohort to form $g_t = \frac{1}{C} \sum_{m \in S_t} g_{t,m}$, and then takes a gradient-type step using this quantity in lieu of the gradient, using server stepsize $\eta > 0$:

$$x_{t+1} = x_t - \eta g_t.$$

The new model is then broadcast to a new cohort in the next communication round, and the process is repeated.

## 5   Warm-up: How to Improve Random Reshuffling

In this section, we provide the intuition behind our complexity improvements through the lens of single-node Random Reshuffling (RR). In particular, when $M = 1$, objective equation 1 recovers the standard empirical-risk

minimization (ERM) problem:

$$\min_{x \in \mathbb{R}^d} \frac{1}{n} \sum_{i=1}^{n} f^i(x).$$

The update of RR for this problem has the form

$$x_t^{i+1} = x_t^i - \gamma \nabla f^{\pi^i}(x_t^i),$$

where we use a permutation $\pi = (\pi^0, \ldots, \pi^{n-1})$ that is randomly sampled at the beginning of epoch $t$. Unrolling this recursion, we get

$$x_t^n = x_t - \gamma \sum_{i=0}^{n-1} \nabla f^{\pi^i}(x_t^i).$$

The key insight is that the gradients evaluated at points $x_t^i$ can be viewed as approximations of the gradients at point $x_t$. If we denote, for simplicity,

$$g_t = \frac{1}{n} \sum_{i=0}^{n-1} \nabla f^{\pi^i}(x_t^i) = \frac{x_t - x_t^n}{\gamma n},$$

then one can show that $g_t \approx \nabla f(x_t)$ whenever $\gamma$ is small. The update of Algorithm 1 becomes much simpler and reduces to

$$\begin{aligned} x_{t+1} = x_t^n + \beta(x_t^n - x_t) &= x_t + (1+\beta)(x_t^n - x_t) \\ &= x_t - (1+\beta)\gamma \sum_{i=0}^{n-1} \nabla f^{\pi^i}(x_t^i) = x_t - \eta g_t, \end{aligned}$$

where $\eta = (1+\beta)\gamma n$. If we imagine for a moment that $g_t$ is indeed a very good approximation of $\nabla f(x_t)$, then the theory of gradient descent suggests that one should use $\eta \sim \frac{1}{L}$, regardless of the value of $\gamma$.

**Complexity improvements.** By following this intuition, we can establish, as special cases of our general theory, several complexity improvements. In strongly convex case, we obtain the $\mathcal{O}\left(\kappa n \log \frac{1}{\varepsilon}\right)$ complexity of the modified Random Reshuffling, which is better than $\mathcal{O}\left(\kappa n + \frac{\sqrt{\kappa n}\sigma_*}{\mu\sqrt{\varepsilon}}\right) \log \frac{1}{\varepsilon}$ of standard Random Reshuffling. In convex case, we our complexity is $\mathcal{O}\left(\frac{Ln}{\varepsilon}\right)$, in contrast to the slower $\mathcal{O}\left(\frac{Ln}{\varepsilon} + \frac{\sqrt{Ln}\sigma_*}{\varepsilon^{3/2}}\right)$ one. Finally, in general non-convex case, we get a bound $\mathcal{O}\left(\frac{Ln}{\varepsilon^2}\right)$, which is better than $\mathcal{O}\left(\frac{Ln}{\varepsilon^2} + \frac{L\sqrt{n}(B+\sqrt{A})}{\varepsilon^3}\right)$, where $A$ and $B$ are defined following Mishchenko et al. (2020) as the constants from the following assumption: $\frac{1}{n} \sum_{i=1}^{n} \|\nabla f_i(x) - \nabla f(x)\|^2 \leq 2A(f(x) - f_*) + B^2$.

## 6 Theory

We now formulate our three main results.

**Theorem 1** (Strongly convex regime)**.** Let Assumption 1 hold, each $f_m^i$ be convex and $f$ be $\mu$-strongly convex. Let $\gamma n \leq \eta \leq \frac{1}{16L}$. Then for iterates $x_t$ generated by Algorithm 1, we have

$$\begin{aligned} \mathbb{E}\left[\|x_T - x_*\|^2\right] \leq &\left(1 - \frac{\eta\mu}{2}\right)^T \|x_0 - x_*\|^2 \\ &+ \frac{5\gamma^2 nL}{\mu}\left(\frac{1}{M}\sum_{m=1}^{M}\sigma_{*,m}^2 + n\sigma_*^2\right) \\ &+ \frac{8\eta}{\mu}\frac{M-C}{C\max\{M-1,1\}}\sigma_*^2. \end{aligned}$$

In the full participation regime, the server stepsize restriction can be relaxed to $\eta \leq \frac{1}{8L}$. Next, we cover the convex regime.

**Theorem 2.** Let Assumption 1 hold, each $f_m^i$ be convex function. Let $\gamma n \leq \eta \leq \frac{1}{16L}$. Let $\hat{x}_T \stackrel{\text{def}}{=} \frac{1}{T} \sum_{t=1}^T x_t$. Then for iterates $x_t$ of Algorithm 1, we have

$$\mathbb{E}[f(\hat{x}_T) - f(x_*)] \leq \frac{5\|x_0 - x_*\|^2}{2\eta T} + 10\eta \frac{M-C}{C \max\{M-1,1\}} \sigma_*^2$$
$$+ 7\gamma^2 nL \left( \frac{1}{M} \sum_{m=1}^M \sigma_{*,m}^2 + n\sigma_*^2 \right)$$

As it can be seen, we get additional source of variance which is proportional to $\eta$ and $\sigma_*^2$. This term means variance of client sampling. Since this sampling of clients have SGD-type structure, we have that variance is proportional to the first order of server-side stepsize.

Finally, we provide guarantees in the non-convex case.

**Theorem 3.** Let Assumption of smoothness hold. Let $\delta_0 = f(x_0) - f_*$ and $\Delta_{*,m} = \frac{1}{n} \sum_{i=1}^n (f_* - f_{*,m}^i)$. Let $\gamma \leq \frac{1}{2nL}$ and $\eta \leq \frac{1}{4L}$. Then for iterates $x_t$ of Algorithm 1, we have

$$\min_{t=0,\ldots,T-1} \mathbb{E}\left[ \|\nabla f(x_t)\|^2 \right] \leq 8L^2 \eta \frac{M-C}{C \max\{M-1,1\}} \Delta_*$$
$$+ 6\gamma^2 nL^3 \left( \frac{1}{M} \sum_{m=1}^M \Delta_{*,m} + n\Delta_* \right)$$
$$+ \frac{4\left(1 + \frac{2L^2\eta^2(M-C)}{C\max\{M-1,1\}} + \frac{3}{2}\eta\gamma^2 n^2 L^3\right)^T}{\eta T} \delta_0.$$

Similarly to analysis in full participation case, we use $\Delta_{*,m}$ and $\Delta_*$ instead of $\sigma_{*,m}^2$ and $\sigma_*^2$, since point of minimizer cannot be defined.

**Client and server stepsizes.** Theorems 1, 2 and 3 suggest that the server can use the large $\mathcal{O}(1/L)$ stepsize, where $L$ is the Lipschitz constant of the gradient of $f$. In all regimes, it is optimal for the client stepsize $\gamma$ to be small, which completely eliminates the second of the three terms in the complexity bounds, which controls the price one pays due to *data heterogeneity*.

**Partial participation.** Notice that if the cohort size is equal to $M$, then $\frac{M-C}{C \max\{1,M-1\}}$ is equal to 0, and this means that the last (third) term in all our complexity results disappears. The last term can thus be interpreted as the price we pay for partial participation. While we can reduce the variance of RR and the client drift by decreasing $\gamma$, we cannot make the variance due to client sampling arbitrary small, since it depends on $\eta$.

**Comparison with existing rates.** In Table 2 we compare our results in the strongly convex and non-convex regimes with selected existing results.

# 7 Benefits of Small Server Stepsize

Our analysis shows that small client stepsizes can control variance. It turns out that using small client stepsizes means that we do not have any benefits from local steps. However, in some cases, our analysis shows that using small server stepsize and large client stepsizes can be beneficial and it means that we gain from using local steps. The advantage of local steps is obtained in case of data reshuffling Mishchenko et al. (2021). Moreover, the goal of learning is not obtaining the best value of the loss function, but the performance of the model. In recent papers, it was shown that large stepsizes are the better option in terms of generalization Smith et al. (2020). Next, we introduce analysis for the case when each $f_i$ is strongly convex.

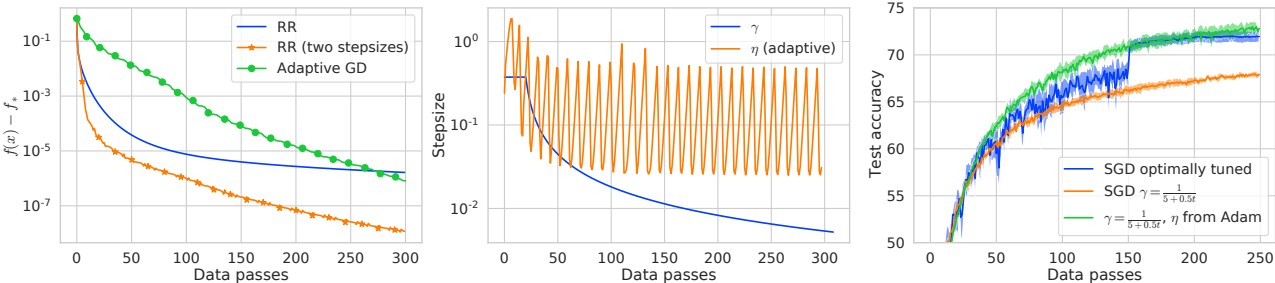

Figure 2: **Left and middle:** We compare running standard Random Reshuffling (RR), adaptive gradient descent (Adaptive GD), and the combination of RR with outer adaptive stepsize (Nastya) (RR (two stepsizes)) on logistic regression. As one can see, the variant with two stepsizes outperforms both of them and does not require more hyper-parameters than RR, and the middle plot shows the exact values of $\gamma$ and $\eta$. **Right:** The right plot shows the training curves of LeNet on CIFAR-10 with minibatch size 1024, where we compare carefully tuned SGD (blue) to poorly tuned SGD (orange) and show that using Adam optimizer with stepsize $10^{-2}$ after each data pass can significantly improve the poorly tuned version.

**Theorem 4.** Assume that all losses $f_{m,i}$ are $L$-smooth and $\mu$-strongly convex. Define $\alpha = \frac{\eta}{\gamma n}$. Let $\gamma \leq \frac{1}{L}$ and $0 \leq \alpha < 1$. Then, for iterates $x_t$ generated by Algorithm 1, we have

$$\mathbb{E}\left[\|x_T - x_*\|^2\right] \leq \left(1 - \alpha + \alpha(1 - \gamma\mu)^n\right)^T \|x_0 - x_*\|^2$$
$$+ \frac{\alpha}{(1-\alpha)(1-(1-\gamma\mu)^n)} \gamma^2 \frac{M-C}{C\max\{M-1,1\}} \sigma_*^2$$
$$+ 2\gamma^3 \sigma_{\mathrm{rad}}^2 \frac{1}{1-(1-\gamma\mu)^n} \sum_{i=0}^{n-1} (1 - \gamma\mu)^i,$$

where $\sigma_{\mathrm{rad}}^2$ is introduced in (Mishchenko et al., 2021) and it corresponds the variance of Random Reshuffling method. The upper bound depends on $\alpha$ in a nonlinear way, so the optimal value of $\alpha$ would often lie somewhere in the interval $(0, 1)$. Furthermore, the last term does not change with $\alpha$, so the optimal value $\alpha^*$ of $\alpha$ is completely determined by the first two terms.

Let us derive optimal $\alpha^*$ under some approximations. In particular, when for ill-conditioned problems where $\mu$ is sufficiently small, it holds $(1 - \gamma\mu)^n \approx 1 - \gamma\mu n$. Ignoring the last term in the upper bound of Theorem 5, which does not affect the value $\alpha^*$, and using $\frac{1}{1-\alpha} \leq 2$ for $\alpha \leq \frac{1}{2}$, we simplify the upper bound to

$$\left(1 - \alpha + \alpha(1 - \gamma\mu n)\right)^T \|x_0 - x_*\|^2$$
$$+ \frac{2\alpha\gamma^2}{1-(1-\gamma\mu n)} \frac{M-C}{C\max\{M-1,1\}} \sigma_*^2$$
$$= (1 - \alpha\gamma\mu n)^T \|x_0 - x_*\|^2 + \frac{2\alpha\gamma}{\mu n} \frac{M-C}{C\max\{M-1,1\}} \sigma_*^2.$$

To have this upper bound smaller than some $\varepsilon \geq 0$, we need to use $\alpha = \mathcal{O}\left(\frac{n\varepsilon C}{\gamma\sigma_*^2}\right)$ and $T = \mathcal{O}(\frac{1}{\alpha\gamma\mu n}\log\frac{1}{\varepsilon})$, where we ignore constants unrelated to $\alpha, \gamma, \varepsilon, \mu$ and $n$. Thus, the server stepsize $\eta = \alpha\gamma n$ should ideally be $\eta = \mathcal{O}\left(\frac{C\varepsilon}{\sigma_*^2}\right)$. In other words, it is better to decrease $\eta$ if only a small subset of clients is used and the variance of client sampling $\frac{M-C}{C\max\{M-1,1\}}\sigma_*^2$ is large.

## 8 Experiments

To showcase the speed-up that can be obtained from the server-side stepsizes, we run a toy experiment in the single-node setup, i.e., we consider standard minimization of a finite-sum. We combine the local passes over the data with the adaptive estimation of smoothness proposed by Malitsky & Mishchenko (2020). We run

our experiment on $\ell_2$-regularized logistic regression with the 'mushrooms' dataset from LibSVM (Chang & Lin, 2011). The results are reported in Figure 2.

We use standard LeNet architecture, which is a 5-layer convolutional neural network, implemented in PyTorch (Paszke et al., 2017) and train them to classify images from the CIFAR-10 dataset (Krizhevsky et al., 2009) with cross-entropy loss. At each iteration, we use a minibatch of size 1024. For the tuned SGD, we start with stepsize 0.2 and divide by 10 at epochs 150 and 200. For the other version, we take SGD with stepsize 0.2 and decrease as $\mathcal{O}(\frac{1}{t})$, where $t$ is the epoch number.

For our method, we treat the full sum of gradients over epoch as an approximation of full gradient and use Adam with stepsize 0.01 to improve this update. We can see from Figure 2 that by applying Adam, we can improve the performance of SGD with decreasing stepsize. At the same time, applying it to the tuned stepsize schedule only made the results much worse, so we do not report that line. This highlights that adaptive outer stepsizes are helpful when the base stepsize $\gamma$ is not chosen well, which is in line with our theory.

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

# A   Appendix

## Contents

# B  Basic Facts and Notation

## B.1  Basic facts

For any two vectors $a, b \in \mathbb{R}^d$ and any $\zeta > 0$,

$$2 \langle a, b \rangle \leq \frac{\|a\|^2}{\zeta} + \zeta \|b\|^2. \tag{7}$$

A consequence of equation 7 is that for any $a, b \in \mathbb{R}^d$, we have

$$\|a + b\|^2 \leq (1 + \zeta) \|a\|^2 + \left(1 + \zeta^{-1}\right) \|b\|^2. \tag{8}$$

Using $\zeta = 1$ specifically yields,

$$\|a + b\|^2 \leq 2\|a\|^2 + 2\|b\|^2. \tag{9}$$

A function $h : \mathbb{R}^d \to \mathbb{R}$ is called $\mu$-convex if for some $\mu \geq 0$ and for all $x, y \in \mathbb{R}^d$, we have

$$h(x) + \langle \nabla h(x), y - x \rangle + \frac{\mu}{2} \|y - x\|^2 \leq h(y). \tag{10}$$

Function $h : \mathbb{R}^d \to \mathbb{R}$ is called $L$-smooth if for some $L \geq 0$ and for all $x, y \in \mathbb{R}^d$, we have

$$\|\nabla h(x) - \nabla h(y)\| \leq L \|x - y\|. \tag{11}$$

A useful consequence of $L$-smoothness is the inequality

$$h(x) \leq h(y) + \langle \nabla h(y), x - y \rangle + \frac{L}{2} \|x - y\|^2, \tag{12}$$

holding for all $x, y \in \mathbb{R}^d$. If $h$ is $L$-smooth and lower bounded by $h_*$, then

$$\|\nabla h(x)\|^2 \leq 2L \left(h(x) - h_*\right). \tag{13}$$

For any convex and $L$-smooth function $h$ it holds

$$\|\nabla h(x) - \nabla h(y)\|^2 \leq 2L D_h(x, y). \tag{14}$$

For a convex function $h \colon \mathbb{R}^d \to \mathbb{R}$ and any vectors $y_1, \ldots, y_n \in \mathbb{R}^d$, Jensen's inequality states that

$$h \left( \frac{1}{n} \sum_{i=1}^n y_i \right) \leq \frac{1}{n} \sum_{i=1}^n h(y_i). \tag{15}$$

Applying this to the squared norm, $h(y) = \|y\|^2$, we get

$$\left\| \frac{1}{n} \sum_{i=1}^n y_i \right\|^2 \leq \frac{1}{n} \sum_{i=1}^n \|y_i\|^2. \tag{16}$$

Simple multiplication on both sides of equation 16 also yields,

$$\left\| \sum_{i=1}^n y_i \right\|^2 \leq n \sum_{i=1}^n \|y_i\|^2. \tag{17}$$

We use the following decomposition that holds for any random variable $X$ with $\mathbb{E}\left[\|X\|^2\right] < +\infty$,

$$\mathbb{E}\left[\|X\|^2\right] = \|\mathbb{E}[X]\|^2 + \mathbb{E}\left[\|X - \mathbb{E}[X]\|^2\right]. \tag{18}$$

We will make use of the particularization of equation 18 to the discrete case: Let $y_1, \ldots, y_n \in \mathbb{R}^d$ be given vectors and let $\bar{y} = \frac{1}{n} \sum_{i=1}^n y_i$ be their average. Then,

$$\frac{1}{n} \sum_{i=1}^n \|y_i\|^2 = \|\bar{y}\|^2 + \frac{1}{n} \sum_{i=1}^n \|y_i - \bar{y}\|^2. \tag{19}$$

## B.2 Notation

We define the variance of the local gradients from their average at a point $x_t$ as

$$\sigma_t^2 \stackrel{\text{def}}{=} \frac{1}{n} \sum_{j=1}^{n} \|\nabla f_j(x_t) - \nabla f(x_t)\|^2.$$

A summary of the notation used is given in Table 3.

Table 3: Summary of notation used.

| Symbol | Description |
|---|---|
| $x_t$ | The iterate used at the start of epoch $t$. |
| $\pi_m$ | A permutation $\pi_m = \left(\pi_m^0, \pi_m^1, \ldots, \pi_m^{n-1}\right)$ of $\{1, 2, \ldots, n\}$, which is resampled every epoch for Random Reshuffling. |
| $\gamma$ | The stepsize used when taking descent steps in an epoch. |
| $x_{t,m}^i$ | The current iterate after $i$ steps in epoch $t$, for $0 \leq i \leq n$. |
| $g_t$ | The sum of gradients used over epoch $t$ such that $x_{t+1} = x_t - \eta g_t$. |
| $\beta$ | The epoch jumping parameter. |
| $\eta$ | The effective epoch stepsize, defined as $\eta \stackrel{\text{def}}{=} \gamma \left(1 + \beta\right) n$. |
| $\sigma_t^2$ | The variance of the individual loss gradients from the average loss at point $x_t$. |
| $L$ | The smoothness constant of $f$ and each $f_m^i$. |
| $\delta_t$ | Functional suboptimality, $\delta_t = f(x_t) - f_*$, where $f_* = \inf_x f(x)$. |

## B.3 Sampling without replacement

We provide the full proof of Lemma 1.

**Lemma.** Let $X_1, \ldots, X_n \in \mathbb{R}^d$ be fixed vectors, $\overline{X} \stackrel{\text{def}}{=} \frac{1}{n} \sum_{i=1}^{n} X_i$ be their average and $\sigma^2 \stackrel{\text{def}}{=} \frac{1}{n} \sum_{i=1}^{n} \|X_i - \overline{X}\|^2$ be the population variance. Fix any $k \in \{1, \ldots, n\}$, let $X_{\pi_1}, \ldots X_{\pi_k}$ be sampled uniformly without replacement from $\{X_1, \ldots, X_n\}$ and $\overline{X}_\pi$ be their average. Then, it holds

$$\mathbb{E}\left[\overline{X}_\pi\right] = \overline{X}, \qquad\qquad \mathbb{E}\left[\left\|\overline{X}_\pi - \overline{X}\right\|^2\right] = \frac{n-k}{k(n-1)}\sigma^2. \qquad (20)$$

*Proof.* The first claim follows by linearity of the expectation and uniformity of the sampling,

$$\mathbb{E}\left[\overline{X}_\pi\right] = \frac{1}{k} \sum_{i=1}^{k} \mathbb{E}\left[X_{\pi_i}\right] = \frac{1}{k} \sum_{i=1}^{k} \overline{X} = \overline{X}.$$

To show the second claim, let us first establish that for any $i \neq j$ it holds $\mathrm{cov}(X_{\pi_i}, X_{\pi_j}) = -\frac{\sigma^2}{n-1}$. Indeed,

$$
\begin{aligned}
\mathrm{cov}(X_{\pi_i}, X_{\pi_j}) &= \mathbb{E}\left[\left\langle X_{\pi_i} - \overline{X}, X_{\pi_j} - \overline{X}\right\rangle\right] = \frac{1}{n(n-1)} \sum_{l=1}^{n} \sum_{m \neq l} \left\langle X_l - \overline{X}, X_m - \overline{X}\right\rangle \\
&= \frac{1}{n(n-1)} \sum_{l=1}^{n} \sum_{m=1}^{n} \left\langle X_l - \overline{X}, X_m - \overline{X}\right\rangle - \frac{1}{n(n-1)} \sum_{l=1}^{n} \left\|X_l - \overline{X}\right\|^2 \\
&= \frac{1}{n(n-1)} \sum_{l=1}^{n} \left\langle X_l - \overline{X}, \sum_{m=1}^{n} (X_m - \overline{X})\right\rangle - \frac{\sigma^2}{n-1} \\
&= -\frac{\sigma^2}{n-1}.
\end{aligned}
$$

Therefore,

$$
\begin{aligned}
\mathbb{E}\left[\left\|\overline{X}_\pi - \overline{X}\right\|^2\right] &= \frac{1}{k^2} \sum_{i=1}^{k} \sum_{j=1}^{k} \mathrm{cov}(X_{\pi_i}, X_{\pi_j}) \\
&= \frac{1}{k^2} \mathbb{E}\left[\sum_{i=1}^{k} \left\|X_{\pi_i} - \overline{X}\right\|^2\right] + \sum_{i=1}^{k} \sum_{j=1, j \neq i}^{n} \mathrm{cov}(X_{\pi_i}, X_{\pi_j}) \\
&= \frac{1}{k^2} \left(k\sigma^2 - k(k-1)\frac{\sigma^2}{n-1}\right) = \frac{n-k}{k(n-1)}\sigma^2.
\end{aligned}
$$

∎

## C   Large Server Stepsize

### C.1   Strongly convex and general convex case

**Lemma 2.** Let Assumption 1 holds and further assume $f$ is $\mu$-strongly convex and each $f_m^i$ is convex. Then

$$
-\frac{1}{Mn} \sum_{m=1}^{M} \sum_{i=0}^{n-1} \left\langle f_m^{\pi_m^i}\left(x_{t,m}^i\right), x_t - x_*\right\rangle \leq -\frac{\mu}{4}\|x_t - x_*\|^2 - \frac{1}{2}\left(f\left(x_t\right) - f\left(x_*\right)\right) + \frac{L}{2Mn} \sum_{m=1}^{M} \sum_{i=0}^{n-1} \left\|x_t - x_{t,m}^i\right\|^2.
$$

*Proof.* We start with the inner product and decompose it using the three-point identity:

$$
\begin{aligned}
\left\langle \nabla f_m^{\pi_m^i}\left(x_{t,m}^i\right), x_t - x_*\right\rangle &= f_m^{\pi_m^i}\left(x_t\right) - f_m^{\pi_m^i}\left(x_*\right) + f_m^{\pi_m^i}\left(x_*\right) - f_m^{\pi_m^i}\left(x_{t,m}^i\right) \\
&\quad + \left\langle \nabla f_m^{\pi_m^i}\left(x_{t,m}^i\right), x_{t,m}^i - x_*\right\rangle - f_m^{\pi_m^i}\left(x_t\right) + f_m^{\pi_m^i}\left(x_{t,m}^i\right) \\
&\quad + \left\langle \nabla f_m^{\pi_m^i}\left(x_{t,m}^i\right), x_t - x_{t,m}^i\right\rangle \\
&= f_m^{\pi_m^i}\left(x_t\right) - f_m^{\pi_m^i}\left(x_*\right) + D_{f_m^{\pi_m^i}}\left(x_*, x_{t,m}^i\right) - D_{f_m^{\pi_m^i}}\left(x_t, x_{t,m}^i\right). \quad (21)
\end{aligned}
$$

Using the representation equation 21, $L$-smoothness and $\mu$-strong convexity we have a bound:

$$-\frac{1}{Mn}\sum_{m=1}^{M}\sum_{i=0}^{n-1}\left\langle f_m^{\pi_m^i}\left(x_{t,m}^i\right), x_t - x_*\right\rangle$$

$$\leq -\frac{1}{Mn}\sum_{m=1}^{M}\sum_{i=0}^{n-1}\left(f_m^{\pi_m^i}(x_t) - f_m^{\pi_m^i}(x_*) + D_{f_m^{\pi_m^i}}\left(x_*, x_{t,m}^i\right) - D_{f_m^{\pi_m^i}}\left(x_t, x_{t,m}^i\right)\right)$$

$$\overset{equation\ 12}{\leq} -(f(x_t) - f(x_*)) - \frac{1}{Mn}\sum_{m=1}^{M}\sum_{i=0}^{n-1}D_{f_m^{\pi_m^i}}\left(x_*, x_{t,m}^i\right) + \frac{L}{2Mn}\sum_{m=1}^{M}\sum_{i=0}^{n-1}\left\|x_t - x_{t,m}^i\right\|^2$$

$$\overset{equation\ 10}{\leq} -\frac{\mu}{4}\|x_t - x_*\|^2 - \frac{1}{2}(f(x_t) - f(x_*)) + \frac{L}{2Mn}\sum_{m=1}^{M}\sum_{i=0}^{n-1}\left\|x_t - x_{t,m}^i\right\|^2.$$

■

**Lemma 3.** Assume that Assumption 1 holds, then

$$\left\|\frac{1}{Cn}\sum_{m\in S_t}\sum_{i=0}^{n-1}\nabla f_m^{\pi_m^i}\left(x_{t,m}^i\right)\right\|^2 \leq 2\frac{L^2}{Cn}\sum_{m\in S_t}\sum_{i=0}^{n-1}\|x_{t,m}^i - x_t\|^2 + 4\left\|\frac{1}{C}\sum_{m\in S_t}\nabla f_m(x_*)\right\|^2 + 8L(f_m(x_t) - f_m(x_*)).$$

*Proof.* We start with Young's inequality. Note that $f_m(x_t) = \frac{1}{n}\sum_{i=0}^{n-1}\nabla f_m^{\pi_m^i}(x_t)$:

$$\left\|\frac{1}{Cn}\sum_{m\in S_t}\sum_{i=0}^{n-1}\nabla f_m^{\pi_m^i}\left(x_{t,m}^i\right)\right\|^2 \overset{equation\ 9}{\leq} 2\left\|\frac{1}{Cn}\sum_{m\in S_t}\sum_{i=0}^{n-1}\left(\nabla f_m^{\pi_m^i}\left(x_{t,m}^i\right) - \nabla f_m^{\pi_m^i}(x_t)\right)\right\|^2 + 2\left\|\frac{1}{C}\sum_{m\in S_t}\nabla f_m(x_t)\right\|^2$$

$$\overset{equation\ 15, equation\ 11}{\leq} 2L^2\frac{1}{Cn}\sum_{m\in S_t}\sum_{i=0}^{n-1}\|x_{t,m}^i - x_t\|^2 + 2\left\|\frac{1}{C}\sum_{m\in S_t}\nabla f_m(x_t)\right\|^2.$$

We use Young's inequality and $L$-smoothness again:

$$\left\|\frac{1}{Cn}\sum_{m\in S_t}\sum_{i=0}^{n-1}\nabla f_m^{\pi_m^i}\left(x_{t,m}^i\right)\right\|^2 \overset{equation\ 15, equation\ 9}{\leq} 2L^2\frac{1}{Cn}\sum_{m\in S_t}\sum_{i=0}^{n-1}\|x_{t,m}^i - x_t\|^2 + 4\left\|\frac{1}{C}\sum_{m\in S_t}\nabla f_m(x_*)\right\|^2$$

$$+ 4\frac{1}{C}\sum_{m\in S_t}\|\nabla f_m(x_t) - \nabla f_m(x_*)\|^2$$

$$\overset{equation\ 14}{\leq} 2L^2\frac{1}{Cn}\sum_{m\in S_t}\sum_{i=0}^{n-1}\|x_{m,t}^i - x_t\|^2 + 4\left\|\frac{1}{C}\sum_{m\in S_t}\nabla f_m(x_*)\right\|^2$$

$$+ 8L\frac{1}{C}\sum_{m\in S_t}(f_m(x_t) - f_m(x_*)).$$

■

**Lemma 4.** Suppose that Algorithm 1 is used and Assumption 1 holds. If $\gamma \leq \frac{1}{2Ln}$, then

$$\frac{1}{Mn}\sum_{m=1}^{M}\sum_{i=0}^{n-1}\mathbb{E}\left[\|x_t - x_{t,m}^i\|^2 | x_t\right] \leq 8\gamma^2 n^2 L(f(x_t) - f(x_*)) + 2\gamma^2 n^2 \frac{1}{M}\sum_{m=1}^{M}\|\nabla f_m(x_*)\|^2 + 2\gamma^2 n\frac{1}{M}\sum_{m=1}^{M}\sigma_{*,m}^2.$$

*Proof.* We start from the definition of $x_{t,m}^i$:

$$\mathbb{E}\left[\left\|x_{t,m}^i - x_t\right\|^2 |x_t\right] = \mathbb{E}\left[\left\|\gamma \sum_{j=0}^{i-1} \nabla f_m^{\pi_m^j}\left(x_{t,m}^j\right)\right\|^2 |x_t\right]$$

$$\overset{equation\ 9}{\leq} 2\gamma^2 \mathbb{E}\left[\left\|\sum_{j=0}^{i-1}\left(\nabla f_m^{\pi_m^j}\left(x_{t,m}^j\right) - \nabla f_m^{\pi_m^j}(x_t)\right)\right\|^2 |x_t\right] + 2\gamma^2 \mathbb{E}\left[\left\|\sum_{j=0}^{i-1} \nabla f_m^{\pi_m^j}(x_t)\right\|^2 |x_t\right]$$

$$\overset{equation\ 15}{\leq} 2\gamma^2 i \sum_{j=0}^{i-1} \mathbb{E}\left[\left\|\nabla f_m^{\pi_m^j}\left(x_{t,m}^j\right) - \nabla f_m^{\pi_m^j}(x_t)\right\|^2 |x_t\right] + 2\gamma^2 \mathbb{E}\left[\left\|\sum_{j=0}^{i-1} \nabla f_m^{\pi_m^j}(x_t)\right\|^2 |x_t\right]$$

$$\overset{equation\ 11}{\leq} 2\gamma^2 L^2 i \sum_{j=0}^{i-1} \mathbb{E}\left[\left\|x_{t,m}^j - x_t\right\|^2 |x_t\right] + 2\gamma^2 \mathbb{E}\left[\left\|\sum_{j=0}^{i-1} \nabla f_m^{\pi_m^j}(x_t)\right\|^2 |x_t\right].$$

Now let us look at the last term. We can apply Lemma 1 and get

$$\mathbb{E}\left[\left\|\sum_{j=0}^{i-1} \nabla f_m^{\pi_m^j}(x_t)\right\|^2 |x_t\right] = i^2 \left\|\nabla f_m(x_t)\right\|^2 + i^2 \mathbb{E}\left[\left\|\frac{1}{i}\sum_{j=0}^{i-1}\left(\nabla f_m^{\pi_m^j}(x_t) - \nabla f_m(x_t)\right)\right\|^2 |x_t\right]$$

$$= i^2 \left\|\nabla f_m(x_t)\right\|^2 + \frac{i(n-i)}{n-1}\sigma_{t,m}^2,$$

where $\sigma_{t,m}^2 \overset{\text{def}}{=} \frac{1}{n}\sum_{i=1}^n \left\|\nabla f_m^i(x_t) - \nabla f_m(x_t)\right\|^2$.

Let us go back:

$$\mathbb{E}\left[\left\|x_{t,m}^i - x_t\right\|^2 |x_t\right] \leq 2\gamma^2 L^2 i \sum_{j=0}^{i-1} \mathbb{E}\left[\left\|x_{t,m}^j - x_t\right\|^2 |x_t\right] + 2\gamma^2 \left(i^2 \left\|\nabla f_m(x_t)\right\|^2 + \frac{i(n-i)}{n-1}\sigma_{t,m}^2\right).$$

Summing the terms leads to

$$\frac{1}{Mn}\sum_{m=1}^M \sum_{i=0}^{n-1} \mathbb{E}\left[\left\|x_{t,m}^i - x_t\right\|^2 |x_t\right] \leq 2\gamma^2 L^2 \frac{1}{Mn}\sum_{m=1}^M \sum_{i=0}^{n-1} i \sum_{j=0}^{i-1} \mathbb{E}\left[\left\|x_{t,m}^j - x_t\right\|^2 |x_t\right]$$

$$+ \frac{2\gamma^2}{Mn}\sum_{m=1}^M \sum_{i=0}^{n-1} i^2 \left\|\nabla f_m(x_t)\right\|^2 + \frac{2\gamma^2}{Mn}\sum_{m=1}^M \sum_{i=0}^{n-1}\frac{i(n-i)\sigma_{t,m}^2}{n-1}$$

$$\leq 2\gamma^2 L^2 \frac{1}{Mn}\sum_{m=1}^M \sum_{i=0}^{n-1} \mathbb{E}\left[\left\|x_{t,m}^i - x_t\right\|^2 |x_t\right] \cdot \frac{n(n-1)}{2}$$

$$+ \frac{2\gamma^2}{Mn}\sum_{m=1}^M \left\|\nabla f_m(x_t)\right\|^2 \cdot \frac{n(n-1)(2n-1)}{6} + \frac{\gamma^2 n(n+1)}{3}\frac{1}{Mn}\sum_{m=1}^M \sigma_{t,m}^2.$$

Choosing $\gamma \leq \frac{1}{2Ln}$, we verify

$$\frac{1}{Mn}\sum_{m=1}^M \sum_{i=0}^{n-1} \mathbb{E}\left[\left\|x_{t,m}^i - x_t\right\|^2 |x_t\right] \leq \frac{4}{3}\left(1 - \gamma^2 L^2 n(n-1)\right)\frac{1}{Mn}\sum_{m=1}^M \sum_{i=0}^{n-1} \mathbb{E}\left[\left\|x_{t,m}^i - x_t\right\|^2 |x_t\right]$$

$$\leq \frac{4\gamma^2}{9}\frac{1}{M}\sum_{m=1}^M \left\|\nabla f_m(x_t)\right\|^2 \cdot (n-1)(2n-1) + \frac{4\gamma^2(n+1)}{9}\frac{1}{M}\sum_{m=1}^M \sigma_{t,m}^2$$

$$\leq \gamma^2 n^2 \frac{1}{M}\sum_{m=1}^M \left\|\nabla f_m(x_t)\right\|^2 + \gamma^2 n \frac{1}{M}\sum_{m=1}^M \sigma_{t,m}^2. \tag{22}$$

Using Young's inequality, we get

$$\frac{1}{Mn} \sum_{m=1}^{M} \sum_{i=0}^{n-1} \mathbb{E}\left[\left\|x_{t,m}^i - x_t\right\|^2 | x_t\right] \overset{equation\ 9, equation\ 19}{\leq} 2\gamma^2 n^2 \frac{1}{M} \sum_{m=1}^{M} \|\nabla f_m(x_t) - \nabla f_m(x_*)\|^2$$

$$+ 2\gamma^2 n^2 \frac{1}{M} \sum_{m=1}^{M} \|\nabla f_m(x_*)\|^2 - \frac{\gamma^2 n}{M} \sum_{m=1}^{M} \|\nabla f_m(x_t)\|^2$$

$$+ 2\gamma^2 n \frac{1}{M} \sum_{m=1}^{M} \frac{1}{n} \sum_{i=0}^{n-1} \mathbb{E}\left[\left\|\nabla f_m^{\pi_m^i}(x_t) - \nabla f_m^{\pi_m^i}(x_*)\right\|^2\right]$$

$$+ 2\gamma^2 n \frac{1}{M} \sum_{m=1}^{M} \frac{1}{n} \sum_{i=0}^{n-1} \mathbb{E}\left[\left\|\nabla f_m^{\pi_m^i}(x_*)\right\|^2\right].$$

Using $L$-smoothness, we obtain

$$\frac{1}{Mn} \sum_{m=1}^{M} \sum_{i=0}^{n-1} \mathbb{E}\left[\left\|x_{t,m}^i - x_t\right\|^2\right] \leq 4\gamma^2 n^2 L \frac{1}{M} \sum_{m=1}^{M} D_{f_m}(x_t, x_*) + 2\gamma^2 n \frac{1}{M} \sum_{m=1}^{M} \sigma_{*,m}^2$$

$$+ 4\gamma^2 n L \frac{1}{M} \sum_{m=1}^{M} \frac{1}{n} \sum_{i=0}^{n-1} D_{f_m^{\pi_m^i}}(x_t, x_*) + 2\gamma^2 n^2 \frac{1}{M} \sum_{m=1}^{M} \|\nabla f_m(x_*)\|^2$$

$$\overset{equation\ 12}{\leq} 8\gamma^2 n^2 L \left(f(x_t) - f(x_*)\right) + 2\gamma^2 n^2 \frac{1}{M} \sum_{m=1}^{M} \|\nabla f_m(x_*)\|^2 + 2\gamma^2 n \frac{1}{M} \sum_{m=1}^{M} \sigma_{*,m}^2.$$

∎

### C.1.1 Proof of Theorem 1

**Theorem.** Assume that Assumption 1 holds and $f$ is $\mu$-strongly convex function. Let $\gamma n \leq \eta \leq \frac{1}{16L}$. Then for iterates $x_t$ generated by Algorithm 1 we have

$$\mathbb{E}\left[\|x_T - x_*\|^2\right] \leq \left(1 - \frac{\eta\mu}{2}\right)^T \|x_0 - x_*\|^2 + \frac{5\gamma^2 nL}{\mu} \frac{1}{M} \sum_{m=1}^{M} \left(\sigma_{*,m}^2 + n\|\nabla f_m(x_*)\|^2\right) + \frac{8\eta}{\mu} \sum_{m=1}^{M} \|\nabla f_m(x_*)\|^2.$$

*Proof.* We start from definition of $x_{t+1}$,

$$\|x_{t+1} - x_*\|^2 = \left\|x_t - \eta \frac{1}{Cn} \sum_{m \in S_t} \sum_{i=0}^{n-1} \nabla f_m^{\pi_m^i}\left(x_{t,m}^i\right) - x_*\right\|^2$$

$$= \|x_t - x_*\|^2 - 2\eta \left\langle \frac{1}{Cn} \sum_{m \in S_t} \sum_{i=0}^{n-1} \nabla f_m^{\pi_m^i}\left(x_{t,m}^i\right), x_t - x_* \right\rangle + \eta^2 \left\|\frac{1}{Cn} \sum_{m \in S_t} \sum_{i=0}^{n-1} \nabla f_m^{\pi_m^i}\left(x_{t,m}^i\right)\right\|^2.$$

Using Lemma 3, we get

$$\|x_{t+1} - x_*\|^2 \leq \|x_t - x_*\|^2 - 2\eta \left\langle \frac{1}{Cn} \sum_{m \in S_t} \sum_{i=0}^{n-1} \nabla f_m^{\pi_m^i}\left(x_{t,m}^i\right), x_t - x_* \right\rangle$$

$$+ \eta^2 \left(2L^2 \frac{1}{Cn} \sum_{m \in S_t} \sum_{i=0}^{n-1} \left\|x_{m,t}^i - x_t\right\|^2 + 4 \left\|\frac{1}{C} \sum_{m \in S_t} \nabla f_m(x_*)\right\|^2 + 8L \frac{1}{C} \sum_{m \in S_t} \left(f_m(x_t) - f_m(x_*)\right)\right).$$

Taking conditional expectation over sampling $S_t$, we get

$$\mathbb{E}_{S_t}\left[\|x_{t+1} - x_*\|^2\right] \leq \|x_t - x_*\|^2 - 2\eta\mathbb{E}_{S_t}\left[\left\langle \frac{1}{Cn}\sum_{m \in S_t}\sum_{i=0}^{n-1}\nabla f_m^{\pi_m^i}\left(x_{t,m}^i\right), x_t - x_*\right\rangle\right]$$

$$+ \eta^2\left(2L^2\mathbb{E}_{S_t}\left[\frac{1}{Cn}\sum_{m \in S_t}\sum_{i=0}^{n-1}\|x_{t,m}^i - x_t\|^2\right] + 4\left\|\frac{1}{C}\sum_{m \in S_t}\nabla f_m(x_*)\right\|^2 + 8L\frac{1}{C}\sum_{m \in S_t}(f_m(x_t) - f_m(x_*))\right)$$

$$\leq \|x_t - x_*\|^2 - 2\eta\left\langle \frac{1}{Mn}\sum_{m=1}^{M}\sum_{i=0}^{n-1}\nabla f_m^{\pi_m^i}\left(x_{t,m}^i\right), x_t - x_*\right\rangle$$

$$+ \eta^2\left(2L^2\frac{1}{Mn}\sum_{m=1}^{M}\sum_{i=0}^{n-1}\|x_{t,m}^i - x_t\|^2 + 4\mathbb{E}_{S_t}\left[\left\|\frac{1}{C}\sum_{m \in S_t}\nabla f_m(x_*)\right\|^2\right] + 8L(f(x_t) - f(x_*))\right)$$

$$\overset{equation\ 1}{\leq} \|x_t - x_*\|^2 - 2\eta\left\langle \frac{1}{Mn}\sum_{m=1}^{M}\sum_{i=0}^{n-1}\nabla f_m^{\pi_m^i}\left(x_{t,m}^i\right), x_t - x_*\right\rangle$$

$$+ \eta^2\left(2L^2\frac{1}{Mn}\sum_{m=1}^{M}\sum_{i=0}^{n-1}\|x_{t,m}^i - x_t\|^2 + 4\frac{M-C}{C\max\{M-1,1\}}\sigma_*^2 + 8L(f(x_t) - f(x_*))\right).$$

Using Lemma 2, we obtain

$$\mathbb{E}_{S_t}\left[\|x_{t+1} - x_*\|^2\right] \leq \|x_t - x_*\|^2 - 2\eta\left(-\frac{\mu}{4}\|x_t - x_*\|^2 - \frac{1}{2}\left(f\left(x_t\right) - f\left(x_*\right)\right) + \frac{L}{2Mn}\sum_{m=1}^{M}\sum_{i=0}^{n-1}\|x_t - x_{t,m}^i\|^2\right)$$

$$+ \eta^2\left(2L^2\frac{1}{Mn}\sum_{m=1}^{M}\sum_{i=0}^{n-1}\|x_{t,m}^i - x_t\|^2 + 4\frac{M-C}{C\max\{M-1,1\}}\sigma_*^2 + 8L(f(x_t) - f(x_*))\right).$$

Rearranging the terms, we obtain:

$$\mathbb{E}_{S_t}\left[\|x_{t+1} - x_*\|^2\right] \leq \|x_t - x_*\|^2\left(1 - \frac{\eta\mu}{2}\right) - \eta\left(1 - 8\eta L\right)\left(f(x_t) - f(x_*)\right)$$

$$+ \eta L\left(1 + 2\eta L\right)\frac{1}{Mn}\sum_{m=1}^{M}\sum_{i=0}^{n-1}\|x_{m,t}^i - x_t\|^2 + 4\eta^2\frac{M-C}{C\max\{M-1,1\}}\sigma_*^2.$$

Using the tower property of conditional expectation and Lemma 4, we get

$$\mathbb{E}\left[\|x_{t+1} - x_*\|^2|x_t\right] \leq \|x_t - x_*\|^2\left(1 - \frac{\eta\mu}{2}\right) + 4\eta^2\frac{M-C}{C\max\{M-1,1\}}\sigma_*^2$$

$$- \eta\left(1 - 8\eta L - (1 + 2\eta L)8\gamma^2 n^2 L^2\right)\left(f(x_t) - f(x_*)\right) \qquad (23)$$

$$+ 2\eta\left(1 + 2\eta L\right)\gamma^2 nL\frac{1}{M}\sum_{m=1}^{M}\left(\sigma_{*,m}^2 + n\|\nabla f_m(x_*)\|^2\right).$$

Taking $\gamma \leq \frac{1}{16nL}$ and $\eta \leq \frac{1}{16L}$, we derive

$$\eta\left(1 - 8\eta L - (1 + 2\eta L)8\gamma^2 n^2 L^2\right)\left(f(x_t) - f(x_*)\right) \geq 0.$$

Taking full expectation yields

$$\mathbb{E}\left[\|x_{t+1} - x_*\|^2\right] \leq \mathbb{E}\left[\|x_t - x_*\|^2\left(1 - \frac{\eta\mu}{2}\right)\right] + \frac{5}{2}\eta\gamma^2 nL\frac{1}{M}\sum_{m=1}^{M}\left(\sigma_{*,m}^2 + n\|\nabla f_m(x_*)\|^2\right) + 4\frac{M-C}{C\max\{M-1,1\}}\sigma_*^2.$$

Unrolling this recursion, we have

$$\mathbb{E}\left[\|x_T - x_*\|^2\right] \leq \left(1 - \frac{\eta\mu}{2}\right)^T\|x_0 - x_*\|^2 + \frac{5\gamma^2 nL}{\mu}\frac{1}{M}\sum_{m=1}^{M}\left(\sigma_{*,m}^2 + n\|\nabla f_m(x_*)\|^2\right) + \frac{8\eta}{\mu}\sum_{m=1}^{M}\|\nabla f_m(x_*)\|^2.$$

$\blacksquare$

## C.2 General convex case

### C.2.1 Proof of Theorem 2

**Theorem.** Let Assumption 1 hold, each $f_m^i$ be convex function. Let $\gamma n \leq \eta \leq \frac{1}{16L}$. Let $\hat{x}_T \stackrel{\text{def}}{=} \frac{1}{T}\sum_{t=1}^{T} x_t$. Then for iterates $x_t$ of Algorithm 1, we have

$$\mathbb{E}[f(\hat{x}_T) - f(x_*)] \leq \frac{5\|x_0 - x_*\|^2}{2\eta T} + 7\gamma^2 nL\left(\frac{1}{M}\sum_{m=1}^{M}\sigma_{*,m}^2 + n\sigma_*^2\right) + 10\eta\frac{M-C}{C\max\{M-1,1\}}\sigma_*^2.$$

*Proof.* We start from equation equation 23 with $\mu = 0$:

$$\mathbb{E}\left[\|x_{t+1} - x_*\|^2 | x_t\right] \leq \|x_t - x_*\|^2 + 4\eta^2\frac{M-C}{C\max\{M-1,1\}}\sigma_*^2$$
$$- \eta\left(1 - 8\eta L - (1 + 2\eta L)8\gamma^2 n^2 L^2\right)(f(x_t) - f(x_*))$$
$$+ 2\eta(1 + 2\eta L)\gamma^2 nL\frac{1}{M}\sum_{m=1}^{M}\left(\sigma_{*,m}^2 + n\|\nabla f_m(x_*)\|^2\right).$$

Using $\gamma n \leq \eta \leq \frac{1}{16L}$, we obtain $-\left(1 - 8\eta L - (1 + 2\eta L)8\gamma^2 n^2 L^2\right) \leq -\frac{4}{10}$

$$\mathbb{E}\left[\|x_{t+1} - x_*\|^2 | x_t\right] \leq \|x_t - x_*\|^2 + 4\eta^2\frac{M-C}{C\max\{M-1,1\}}\sigma_*^2 - \frac{4\eta}{10}(f(x_t) - f(x_*))$$
$$+ \frac{5}{2}\eta\gamma^2 nL\frac{1}{M}\sum_{m=1}^{M}\left(\sigma_{*,m}^2 + n\|\nabla f_m(x_*)\|^2\right).$$

Taking full expectation, we get

$$\mathbb{E}\left[\|x_{t+1} - x_*\|^2\right] \leq \mathbb{E}\left[\|x_t - x_*\|^2\right] + 4\eta^2\frac{M-C}{C\max\{M-1,1\}}\sigma_*^2 - \frac{4\eta}{10}\mathbb{E}\left[(f(x_t) - f(x_*))\right]$$
$$+ \frac{5}{2}\eta\gamma^2 nL\frac{1}{M}\sum_{m=1}^{M}\left(\sigma_{*,m}^2 + n\|\nabla f_m(x_*)\|^2\right).$$

Rearranging the terms leads us to

$$\frac{4\eta}{10}\mathbb{E}\left[(f(x_t) - f(x_*))\right] \leq \mathbb{E}\left[\|x_t - x_*\|^2\right] - \mathbb{E}\left[\|x_{t+1} - x_*\|^2\right] + 4\eta^2\frac{M-C}{C\max\{M-1,1\}}\sigma_*^2$$
$$+ \frac{5}{2}\eta\gamma^2 nL\frac{1}{M}\sum_{m=1}^{M}\left(\sigma_{*,m}^2 + n\|\nabla f_m(x_*)\|^2\right).$$

Averaging from $0$ to $T-1$, we get

$$\frac{4\eta}{10}\frac{1}{T}\sum_{t=0}^{T-1}\left[(f(x_t) - f(x_*))\right] \leq \frac{1}{T}\sum_{t=0}^{T-1}\left(\mathbb{E}\left[\|x_t - x_*\|^2\right] - \mathbb{E}\left[\|x_{t+1} - x_*\|^2\right]\right) + 4\eta^2\frac{M-C}{C\max\{M-1,1\}}\sigma_*^2$$
$$+ \frac{5}{2}\eta\gamma^2 nL\frac{1}{M}\sum_{m=1}^{M}\left(\sigma_{*,m}^2 + n\|\nabla f_m(x_*)\|^2\right)$$
$$\leq \frac{1}{T}\left(\mathbb{E}\left[\|x_0 - x_*\|^2\right] - \mathbb{E}\left[\|x_T - x_*\|^2\right]\right) + 4\eta^2\frac{M-C}{C\max\{M-1,1\}}\sigma_*^2$$
$$+ \frac{5}{2}\eta\gamma^2 nL\frac{1}{M}\sum_{m=1}^{M}\left(\sigma_{*,m}^2 + n\|\nabla f_m(x_*)\|^2\right).$$

Using Jensen inequality equation 15, we have

$$\mathbb{E}[f(\hat{x}_T) - f(x_*)] \leq \frac{5\left\|x_0 - x_*\right\|^2}{2\eta T} + 7\gamma^2 nL\left(\frac{1}{M}\sum_{m=1}^{M}\sigma_{*,m}^2 + n\sigma_*^2\right) + 10\eta\frac{M-C}{C\max\{M-1,1\}}\sigma_*^2.$$

∎

### C.3 General non-convex case

Finally, we provide guarantees in the non-convex case.

**Lemma 5.** Assume that Assumption 1. For uniform sampling of cohort $S_t$ we have

$$\frac{L}{2}\eta^2\mathbb{E}_{S_t}\left[\left\|\frac{1}{Cn}\sum_{m\in S_t}\sum_{i=0}^{n-1}\nabla f_m^{\pi_m^i}(x_{t,m}^i)\right\|^2\right] \leq L^3\eta^2\mathbb{E}_{S_t}\left[\frac{1}{Mn}\sum_{m=1}^{M}\sum_{i=0}^{n-1}\left\|x_{t,m}^i - x_t\right\|^2\right] + L\eta^2\|\nabla f(x_t)\|^2$$

$$+ L\eta^2\frac{M-C}{C\max\{M-1,1\}}\left(2L(f(x_t) - f(x_*)) + 2L\Delta_*\right).$$

*Proof.* We start from Young's inequality and then we use Jensen's inequality:

$$\frac{L}{2}\eta^2\mathbb{E}_{S_t}\left[\left\|\frac{1}{Cn}\sum_{m\in S_t}\sum_{i=0}^{n-1}\nabla f_m^{\pi_m^i}(x_{t,m}^i)\right\|^2\right] \overset{equation\ 9}{\leq} L\eta^2\mathbb{E}_{S_t}\left[\left\|\frac{1}{Cn}\sum_{m\in S_t}\sum_{i=0}^{n-1}\left(\nabla f_m^{\pi_m^i}(x_{t,m}^i) - \nabla f_m^{\pi_m^i}(x_t)\right)\right\|^2\right]$$

$$+ L\eta^2\mathbb{E}_{S_t}\left[\left\|\frac{1}{Cn}\sum_{m\in S_t}\sum_{i=0}^{n-1}\nabla f_m^{\pi_m^i}(x_t)\right\|^2\right]$$

$$\overset{equation\ 15}{\leq} L\eta^2\mathbb{E}_{S_t}\left[\frac{1}{Cn}\sum_{m\in S_t}\sum_{i=0}^{n-1}\left\|\nabla f_m^{\pi_m^i}(x_{t,m}^i) - \nabla f_m^{\pi_m^i}(x_t)\right\|^2\right]$$

$$+ L\eta^2\mathbb{E}_{S_t}\left[\left\|\frac{1}{Cn}\sum_{m\in S_t}\sum_{i=0}^{n-1}\nabla f_m^{\pi_m^i}(x_t)\right\|^2\right]$$

$$\overset{equation\ 11}{\leq} L^3\eta^2\mathbb{E}_{S_t}\left[\frac{1}{Cn}\sum_{m\in S_t}\sum_{i=0}^{n-1}\left\|x_{t,m}^i - x_t\right\|^2\right]$$

$$+ L\eta^2\mathbb{E}_{S_t}\left[\left\|\frac{1}{Cn}\sum_{m\in S_t}\sum_{i=0}^{n-1}\nabla f_m^{\pi_m^i}(x_t)\right\|^2\right].$$

Taking expectations and using Lemma 1 we get

$$\frac{L}{2}\eta^2\mathbb{E}_{S_t}\left[\left\|\frac{1}{Cn}\sum_{m\in S_t}\sum_{i=0}^{n-1}\nabla f_m^{\pi_m^i}(x_{t,m}^i)\right\|^2\right] \overset{equation\ 6}{\leq} L^3\eta^2\frac{1}{Mn}\sum_{m=1}^{M}\sum_{i=0}^{n-1}\left\|x_{t,m}^i - x_t\right\|^2$$

$$+ L\eta^2\left(\nabla f(x_t) + \frac{M-C}{C\max\{M-1,1\}}\sigma_t^2\right)$$

Next, we follow steps of Proposition 2 from Mishchenko et al. (2020). Using the definition $\sigma_t^2 = \frac{1}{M} \sum_{m=1}^{M} \|\nabla f_m(x_t) - \nabla f(x_t)\|^2$ we obtain

$$
\begin{aligned}
\frac{L}{2} \eta^2 \mathbb{E}_{S_t} \left[ \left\| \frac{1}{Cn} \sum_{m \in S_t} \sum_{i=0}^{n-1} \nabla f_m^{\pi_m^i}(x_{t,m}^i) \right\|^2 \right] &\leq L^3 \eta^2 \frac{1}{Mn} \sum_{m=1}^{M} \sum_{i=0}^{n-1} \|x_{t,m}^i - x_t\|^2 \\
&\quad + L\eta^2 \left( \|\nabla f(x_t)\|^2 + \frac{M-C}{C \max\{M-1,1\}} \frac{1}{M} \sum_{m=1}^{M} \|\nabla f_m(x_t) - \nabla f(x_t)\|^2 \right) \\
&\overset{equation\ 19}{=} L^3 \eta^2 \frac{1}{Mn} \sum_{m=1}^{M} \sum_{i=0}^{n-1} \|x_{t,m}^i - x_t\|^2 \\
&\quad + L\eta^2 \left( \|\nabla f(x_t)\|^2 + \frac{M-C}{C \max\{M-1,1\}} \left( \frac{1}{M} \sum_{m=1}^{M} \|\nabla f_m(x_t)\|^2 - \|\nabla f(x_t)\|^2 \right) \right) \\
&\leq L^3 \eta^2 \frac{1}{Mn} \sum_{m=1}^{M} \sum_{i=0}^{n-1} \|x_{t,m}^i - x_t\|^2 \\
&\quad + L\eta^2 \left( \|\nabla f(x_t)\|^2 + \frac{M-C}{C \max\{M-1,1\}} \frac{1}{M} \sum_{m=1}^{M} \|\nabla f_m(x_t)\|^2 \right) \\
&\leq L^3 \eta^2 \frac{1}{Mn} \sum_{m=1}^{M} \sum_{i=0}^{n-1} \|x_{t,m}^i - x_t\|^2 + L\eta^2 \|\nabla f(x_t)\|^2 \\
&\quad + L\eta^2 \frac{M-C}{C \max\{M-1,1\}} \left( 2L(f(x_t) - f_*) + 2L \left( f_* - \frac{1}{M} \sum_{m=1}^{M} f_{*,m} \right) \right).
\end{aligned}
$$

Finally, we get

$$
\begin{aligned}
\frac{L}{2} \eta^2 \mathbb{E}_{S_t} \left[ \left\| \frac{1}{Cn} \sum_{m \in S_t} \sum_{i=0}^{n-1} \nabla f_m^{\pi_m^i}(x_{t,m}^i) \right\|^2 \right] &\leq L^3 \eta^2 \mathbb{E}_{S_t} \left[ \frac{1}{Mn} \sum_{m=1}^{M} \sum_{i=0}^{n-1} \|x_{t,m}^i - x_t\|^2 \right] + L\eta^2 \|\nabla f(x_t)\|^2 \\
&\quad + L\eta^2 \frac{M-C}{C \max\{M-1,1\}} \left( 2L(f(x_t) - f(x_*)) + 2L\Delta_* \right).
\end{aligned}
$$

∎

**Lemma 6.** Suppose that Algorithm 1 is used and Assumption 1 holds. If $\gamma \leq \frac{1}{2Ln}$, then

$$
\frac{1}{Mn} \sum_{m=1}^{M} \sum_{i=0}^{n-1} \mathbb{E} \left[ \|x_t - x_{t,m}^i\|^2 \mid x_t \right] \leq 4\gamma^2 n^2 L (f(x_t) - f_*) + 2\gamma^2 n^2 L\Delta_* + 2\gamma^2 nL \frac{1}{M} \sum_{m=1}^{M} \Delta_{*,m}.
$$

*Proof.* We start from equation equation 22. It is proved in section C.1 but it is not required convexity:

$$
\frac{1}{Mn} \sum_{m=1}^{M} \sum_{i=0}^{n-1} \mathbb{E} \left[ \|x_{t,m}^i - x_t\|^2 \mid x_t \right] \leq \gamma^2 n^2 \frac{1}{M} \sum_{m=1}^{M} \|\nabla f_m(x_t)\|^2 + \gamma^2 n \frac{1}{M} \sum_{m=1}^{M} \sigma_{t,m}^2.
$$

Using $L$-smoothness, we get

$$\frac{1}{Mn} \sum_{m=1}^{M} \sum_{i=0}^{n-1} \mathbb{E}\left[\left\|x_{t,m}^i - x_t\right\|^2 |x_t\right] \leq 2\gamma^2 n^2 L \frac{1}{M} \sum_{m=1}^{M} (f_m(x_t) - f_{*,m}) + 2\gamma^2 nL \frac{1}{M} \sum_{m=1}^{M} \frac{1}{n} \sum_{i=0}^{n-1} (f_m^i(x_t) - f_{*,m}^i)$$

$$\leq 2\gamma^2 n^2 L \frac{1}{M} \sum_{m=1}^{M} (f_m(x_t) - f_*) + 2\gamma^2 n^2 L \frac{1}{M} \sum_{m=1}^{M} (f_* - f_{*,m})$$

$$+ 2\gamma^2 nL \frac{1}{M} \sum_{m=1}^{M} \frac{1}{n} \sum_{i=0}^{n-1} (f_m^i(x_t) - f_*) + 2\gamma^2 nL \frac{1}{M} \sum_{m=1}^{M} \frac{1}{n} \sum_{i=0}^{n-1} (f_* - f_{*,m}^i)$$

$$\leq 4L\gamma^2 n^2 (f(x_t) - f_*) + 2\gamma^2 n^2 L\Delta_* + 2\gamma^2 nL \frac{1}{M} \sum_{m=1}^{M} \Delta_{*,m}.$$

■

**Lemma 7.** Suppose that there exists constants $a, b, c \geq 0$ and nonnegative sequences $(s_t)_{t=0}^T, (q_t)_{t=0}^T$ such that for any $t \in \{0, 1, \ldots, T\}$

$$s_{t+1} \leq (1+a)\, s_t - bq_t + c. \tag{24}$$

Then if $a > 0$ we have,

$$\min_{t=0,\ldots,T-1} q_t \leq \frac{(1+a)^T}{bT} s_0 + \frac{c}{b}. \tag{25}$$

And if $a = 0$ we have,

$$\frac{1}{T} \sum_{t=0}^{T-1} q_t \leq \frac{s_0}{bT} + \frac{c}{b}. \tag{26}$$

*Proof.* The first part of the proof (for $a > 0$) is a distillation of the recursion solution in Lemma 2 of (?) and we closely follow their proof. Let $w_{-1} = w_0 > 0$ be arbitrary. Define

$$w_t \stackrel{\text{def}}{=} \frac{w_0}{(1+a)^t}.$$

Note that $w_t (1 + a) = w_{t-1}$. Multiplying both sides of equation 24 by $w_t$,

$$w_t s_{t+1} \leq (1+a)\, w_t s_t - bw_t q_t + cw_t$$
$$= w_{t-1} s_t - bw_t q_t + cw_t.$$

Rearranging,

$$bw_t q_t \leq w_{t-1} s_t - w_t s_{t+1} + cw_t.$$

Summing up as $t$ varies from 0 to $T - 1$ and noting that the sum telescopes,

$$\sum_{t=0}^{T-1} bw_t q_t \leq \sum_{t=0}^{T-1} (w_{t-1} s_t - w_t s_{t+1}) + c \sum_{t=0}^{T-1} w_t = w_0 s_0 - w_{T-1} s_T + c \sum_{t=0}^{T-1} w_t \leq w_0 s_0 + c \sum_{t=0}^{T-1} w_t.$$

Let $W_T = \sum_{t=0}^{T-1} w_t$. Dividing both sides by $W_T$ we have,

$$\frac{1}{W_T} \sum_{t=0}^{T-1} bw_t q_t \leq \frac{w_0 s_0}{W_T} + c. \tag{27}$$

We now separate the proof into two cases:

- **If** $a > 0$: Note that the left-hand side of equation 27 satisfies

$$b \min_{t=0,\ldots,T-1} q_t \leq \frac{1}{W_T} \sum_{t=0}^{T-1} b w_t q_t. \tag{28}$$

And for the right hand-side of equation 27 we have,

$$W_T = \sum_{t=0}^{T-1} w_t \geq T \min_{t=0,\ldots,T-1} w_t = T w_{T-1} \geq T w_T = \frac{T w_0}{(1+a)^T}. \tag{29}$$

Substituting with equation 29 in equation 28 and dividing both sides by $b$ we get,

$$\min_{t=0,\ldots,T-1} q_t \leq \frac{(1+a)^T}{bT} s_0 + \frac{c}{b}.$$

- **If** $a = 0$: then $w_t = w_0$ for all $t$ and hence $w_T = T$, then equation 28 is equivalent to

$$\frac{1}{T} \sum_{t=0}^{T-1} b q_t \leq \frac{s_0}{T} + c.$$

Dividing both sides by $b$ yields the lemma's claim. ∎

### C.3.1 Proof of Theorem 3

**Theorem.** Let Assumption of smoothness hold. Let $\delta_0 = f(x_0) - f_*$ and $\Delta_{*,m} = \frac{1}{n} \sum_{i=1}^{n} (f_* - f_{*,m}^i)$. Let $\gamma \leq \frac{1}{2nL}$ and $\eta \leq \frac{1}{4L}$. Then for iterates $x_t$ of Algorithm 1, we have

$$\min_{t=0,\ldots,T-1} \mathbb{E}\left[\|\nabla f(x_t)\|^2\right] \leq 8 L^2 \eta \frac{M-C}{C \max\{M-1,1\}} \Delta_*$$

$$+ 6\gamma^2 n L^3 \left(\frac{1}{M} \sum_{m=1}^{M} \Delta_{*,m} + n\Delta_*\right) + \frac{4\left(1 + \frac{2L^2\eta^2(M-C)}{C\max\{M-1,1\}} + \frac{3}{2}\eta\gamma^2 n^2 L^3\right)^T}{\eta T} \delta_0.$$

*Proof.* We start from $L$-smoothness equation 12:

$$f(x_{t+1}) \overset{\text{equation } 12}{\leq} f(x_t) + \langle \nabla f(x_t), x_{t+1} - x_t \rangle + \frac{L}{2} \|x_{t+1} - x_t\|^2$$

$$= f(x_t) - \left\langle \nabla f(x_t), \eta \frac{1}{Cn} \sum_{m \in S_t} \sum_{i=0}^{n-1} \nabla f_m^{\pi_m^i}(x_{t,m}^i) \right\rangle + \frac{L}{2} \left\| \eta \frac{1}{Cn} \sum_{m \in S_t} \sum_{n=0}^{n-1} \nabla f_m^{\pi_m^i}(x_{t,m}^i) \right\|^2$$

$$= f(x_t) - \eta \left\langle \nabla f(x_t), \frac{1}{Cn} \sum_{m \in S_t} \sum_{i=0}^{n-1} \nabla f_m^{\pi_m^i}(x_{t,m}^i) \right\rangle + \frac{L}{2} \eta^2 \left\| \frac{1}{Cn} \sum_{m \in S_t} \sum_{n=0}^{n-1} \nabla f_m^{\pi_m^i}(x_{t,m}^i) \right\|^2.$$

Taking conditional expectation over cohort $S_t$, we get

$$\mathbb{E}_{S_t}[f(x_{t+1})] \leq f(x_t) - \eta \mathbb{E}_{S_t}\left[\left\langle \nabla f(x_t), \frac{1}{Cn} \sum_{m \in S_t} \sum_{i=0}^{n-1} \nabla f_m^{\pi_m^i}(x_{t,m}^i) \right\rangle\right] + \frac{L}{2} \eta^2 \mathbb{E}_{S_t}\left[\left\| \frac{1}{Cn} \sum_{m \in S_t} \sum_{n=0}^{n-1} \nabla f_m^{\pi_m^i}(x_{t,m}^i) \right\|^2\right]$$

$$= f(x_t) - \eta \left\langle \nabla f(x_t), \frac{1}{Mn} \sum_{m=1}^{M} \sum_{i=0}^{n-1} \nabla f_m^{\pi_m^i}(x_{t,m}^i) \right\rangle + \frac{L}{2} \eta^2 \mathbb{E}_{S_t}\left[\left\| \frac{1}{Cn} \sum_{m \in S_t} \sum_{n=0}^{n-1} \nabla f_m^{\pi_m^i}(x_{t,m}^i) \right\|^2\right].$$

Using $2 \langle a, b \rangle = \|a + b\|^2 - \|a\|^2 - \|b\|^2$, we have

$$
\mathbb{E}_{S_t} [f(x_{t+1})] = f(x_t) + \frac{L}{2}\eta^2 \mathbb{E}_{S_t} \left[ \left\| \frac{1}{Cn} \sum_{m \in S_t} \sum_{n=0}^{n-1} \nabla f_m^{\pi_m^i} \left(x_{t,m}^i\right) \right\|^2 \right]
$$
$$
- \left( \frac{\eta}{2}\|\nabla f(x_t)\|^2 + \frac{\eta}{2} \left\| \frac{1}{Mn} \sum_{m=1}^{M} \sum_{i=0}^{n-1} \nabla f_m^{\pi_m^i} \left(x_{t,m}^i\right) \right\|^2 \right) + \frac{\eta}{2} \left\| \nabla f(x_t) - \frac{1}{Mn} \sum_{m=1}^{M} \sum_{i=0}^{n-1} \nabla f_m^{\pi_m^i} \left(x_{t,m}^i\right) \right\|^2
$$
$$
\leq f(x_t) + \frac{L}{2}\eta^2 \mathbb{E}_{S_t} \left[ \left\| \frac{1}{Cn} \sum_{m \in S_t} \sum_{n=0}^{n-1} \nabla f_m^{\pi_m^i} \left(x_{t,m}^i\right) \right\|^2 \right]
$$
$$
- \left( \frac{\eta}{2}\|\nabla f(x_t)\|^2 + \frac{\eta}{2} \left\| \frac{1}{Mn} \sum_{m=1}^{M} \sum_{i=0}^{n-1} \nabla f_m^{\pi_m^i} \left(x_{t,m}^i\right) \right\|^2 \right) + \frac{\eta}{2} \left\| \frac{1}{Mn} \sum_{m=1}^{M} \sum_{i=0}^{n-1} \left( \nabla f_m^{\pi_m^i} \left(x_{t,m}^i\right) - \nabla f_m^{\pi_m^i} \left(x_t\right) \right) \right\|^2.
$$

Using $L$-smoothness, we get

$$
\mathbb{E}_{S_t} [f(x_{t+1})] \leq f(x_t) + \frac{L}{2}\eta^2 \mathbb{E}_{S_t} \left[ \left\| \frac{1}{Cn} \sum_{m \in S_t} \sum_{n=0}^{n-1} \nabla f_m^{\pi_m^i} \left(x_{t,m}^i\right) \right\|^2 \right]
$$
$$
- \frac{\eta}{2}\|\nabla f(x_t)\|^2 + \frac{\eta}{2}L^2 \frac{1}{Mn} \sum_{m=1}^{M} \sum_{i=0}^{n-1} \left\| x_{t,m}^i - x_t \right\|^2.
$$

Utilizing Lemma 5 and taking conditional expectation, we get

$$
\mathbb{E}\left[f(x_{t+1})|x_t\right] \leq f(x_t) + L^3 \eta^2 \frac{1}{Mn} \sum_{m=1}^{M} \sum_{i=0}^{n-1} \mathbb{E}\left[ \left\| x_{t,m}^i - x_t \right\|^2 |x_t \right] + L\eta^2 \|\nabla f(x_t)\|^2
$$
$$
+ L\eta^2 \frac{M - C}{C \max\{M - 1, 1\}} \left(2L(f(x_t) - f(x_*)) + 2L\Delta_*\right)
$$
$$
- \frac{\eta}{2}\|\nabla f(x_t)\|^2 + \frac{\eta}{2}L^2 \frac{1}{Mn} \sum_{m=1}^{M} \sum_{i=0}^{n-1} \mathbb{E}\left[ \left\| x_{t,m}^i - x_t \right\|^2 |x_t \right]
$$
$$
\leq f(x_t) + \frac{3}{4}\eta L^2 \frac{1}{Mn} \sum_{m=1}^{M} \sum_{i=0}^{n-1} \mathbb{E}\left[ \left\| x_{t,m}^i - x_t \right\|^2 |x_t \right] - \frac{\eta}{4}\|\nabla f(x_t)\|^2
$$
$$
+ L\eta^2 \frac{M - C}{C \max\{M - 1, 1\}} \left(2L(f(x_t) - f(x_*)) + 2L\Delta_*\right).
$$

Applying Lemma 6 and using $\eta \leq \frac{1}{4L}$ we get

$$
\mathbb{E}\left[f(x_{t+1})|x_t\right] \leq f(x_t) + L\eta^2 \frac{M - C}{C \max\{M - 1, 1\}} \left(2L(f(x_t) - f(x_*)) + 2L\Delta_*\right)
$$
$$
- \frac{\eta}{4}\|\nabla f(x_t)\|^2 + \frac{3\eta}{4}L^2 \left(4L\gamma^2 n^2(f(x_t) - f_*) + 2\gamma^2 n^2 L\Delta_* + 2\gamma^2 nL \frac{1}{M} \sum_{m=1}^{M} \Delta_{*,m}\right).
$$

Substracting $f_*$ from both side leads to

$$
\mathbb{E}\left[f(x_{t+1})|x_t\right] - f_* \leq f(x_t) - f_* + L\eta^2 \frac{M - C}{C \max\{M - 1, 1\}} \left(2L(f(x_t) - f(x_*)) + 2L\Delta_*\right)
$$
$$
- \frac{\eta}{4}\|\nabla f(x_t)\|^2 + \frac{3\eta}{4}L^2 \left(4L\gamma^2 n^2(f(x_t) - f_*) + 2\gamma^2 n^2 L\Delta_* + 2\gamma^2 nL \frac{1}{M} \sum_{m=1}^{M} \Delta_{*,m}\right).
$$

Taking full expectation, we have

$$\mathbb{E}\left[\delta_{t+1}\right] \leq \left(1 + \frac{2L^2\eta^2}{C} + \frac{3}{2}\eta\gamma^2 n^2 L^3\right)\mathbb{E}\left[\delta_t\right] - \frac{\eta}{4}\mathbb{E}\left[\|\nabla f(x_t)\|^2\right]$$

$$+ 2L^2\eta^2 \frac{M-C}{C\max\{M-1,1\}}\Delta_* + \frac{3}{2}\eta\gamma^2 n^2 L^3 \Delta_* + \frac{3}{2}\eta\gamma^2 nL^3 \frac{1}{M}\sum_{m=1}^{M}\Delta_{*,m}.$$

Applying Lemma 7 from Mishchenko et al. (2020), we get

$$\min_{t=0,\ldots,T-1}\mathbb{E}\left[\|\nabla f(x_t)\|^2\right] \leq \frac{4\left(1 + \frac{2L^2\eta^2}{C} + \frac{3}{2}\eta\gamma^2 n^2 L^3\right)^T}{\eta T}\delta_0 + 6\gamma^2 nL^3\left(\frac{1}{M}\sum_{m=1}^{M}\Delta_{*,m} + n\Delta_*\right)$$

$$+ 8L^2\eta\frac{M-C}{C\max\{M-1,1\}}\Delta_*.$$

∎

# D  Small Server Stepsize

In this section, we present a result when it is useful to pull back the last iterates of local passes. In particular, we show that one can reduce the variance of FedAvg with uniform partial participation.

**Theorem 5.** Assume that all losses $f_{m,i}$ are $L$-smooth and $\mu$-strongly convex. Define $\alpha = \frac{\eta}{\gamma n}$. Let $\gamma \leq \frac{1}{L}$ and $0 \leq \alpha < 1$. Then, for iterates $x_t$ generated by Algorithm 1, we have

$$\mathbb{E}\left[\|x_T - x_*\|^2\right] \leq (1 - \alpha + \alpha(1 - \gamma\mu)^n)^T\|x_0 - x_*\|^2$$

$$+ \frac{\alpha}{(1-\alpha)(1-(1-\gamma\mu)^n)}\gamma^2\frac{M-C}{C\max\{M-1,1\}}\sigma_*^2 + 2\gamma^3\sigma_{\mathrm{rad}}^2\frac{1}{1-(1-\gamma\mu)^n}\sum_{i=0}^{n-1}(1-\gamma\mu)^i.$$

*Proof.* Let us denote $f_{S_t} = \frac{1}{C}\sum_{m\in S_t} f_m$. We start by rewriting the distance to the optimum in the following way:

$$x_{t+1} - x_* = (1-\alpha)x_t + \alpha x_t^n - x_*$$

$$= (1-\alpha)x_t + \alpha x_t^n - (1-\alpha)\left(x_* + \frac{\alpha}{1-\alpha}\gamma n\nabla f_{S_t}(x_*)\right) - \alpha(x_* - \gamma n\nabla f_{S_t}(x_*)).$$

Therefore, by convexity of the squared norm,

$$\|x_{t+1} - x_*\|^2 \leq (1-\alpha)\|x_t - \left(x_* + \frac{\alpha}{1-\alpha}\gamma n\nabla f_{S_t}(x_*)\right)\|^2 + \alpha\|x_t^n - (x_* - \gamma n\nabla f_{S_t}(x_*))\|^2.$$

We bound the two terms in the right-hand side separately. For the first term, it suffices to take expectation over the sampling of client cohort $S_t$,

$$\mathbb{E}_{S_t}\|x_t - \left(x_* + \frac{\alpha}{1-\alpha}\gamma n\nabla f_{S_t}(x_*)\right)\|^2 \stackrel{equation\ 18}{=} \|x_t - x_*\|^2 + \frac{\alpha^2}{(1-\alpha)^2}\gamma^2 n^2 \mathbb{E}_{S_t}\|\nabla f_{S_t}(x_*)\|^2$$

$$= \|x_t - x_*\|^2 + \frac{\alpha^2}{(1-\alpha)^2}\gamma^2 n^2 \frac{M-C}{C\max\{M-1,1\}}\sigma_*^2.$$

For the second term, we use the results of prior work on convergence of RR that gives

$$\|x_t^n - (x_* - \gamma\nabla f_{S_t}(x_*))\|^2 \leq (1-\gamma\mu)^n\|x_t - x_*\|^2 + 2\gamma^3\sigma_{\mathrm{rad}}^2\sum_{i=0}^{n-1}(1-\gamma\mu)^i,$$

where, as shown by Mishchenko et al. (2021), $\sigma_{\mathrm{rad}} \geq 0$ is some constant satisfying

$$\sigma_{\mathrm{rad}}^2 \leq L \sum_{m=1}^{M} (n^2 \|\nabla f_m(x_*)\|^2 + \frac{n}{4}\sigma_{*,m}^2).$$

Notice that the upper bound depends on $\alpha$ in a nonlinear way, so the optimal value of $\alpha$ would often lie somewhere in the interval $(0, 1)$. Recurrence $a_{t+1} \leq (1 - \rho)a_t + c$ implies by induction $a_t \leq (1 - \rho)^t a_0 + \frac{c}{\rho}$, so by propagating the bound above to $x_0$, we obtain

$$\mathbb{E}\|x_t - x_*\|^2 \leq (1 - \alpha + \alpha(1 - \gamma\mu)^n)^t \|x_0 - x_*\|^2 + \frac{\alpha}{(1 - \alpha)(1 - (1 - \gamma\mu)^n)}\gamma^2 \frac{M - C}{C\max\{M - 1, 1\}}\sigma_*^2$$

$$+ 2\gamma^3\sigma_{\mathrm{rad}}^2 \frac{1}{1 - (1 - \gamma\mu)^n} \sum_{i=0}^{n-1}(1 - \gamma\mu)^i.$$

Notice that the last term does not change with $\alpha$, so its optimal value is completely determined by the first two terms. ∎

