# OpenReview forum: "Server-Side Stepsizes and Sampling Without Replacement Provably Help in Federated Optimization"
_TMLR — Rejected by TMLR_

### Review · Reviewer_abhw · 2022-09-28

**Summary Of Contributions:**

This paper studies a generalization of the FedAvg algorithm that employs client and server learning rates, along with random reshuffling on clients. The paper's primary contributions are providing convergence bounds for this method (with partial participation of clients) in strongly convex, convex, and non-convex settings. Notably, the use of random reshuffling does improve convergence over FedAvg without random reshuffling. Using this theory, the authors provide some insight into when it may be advantageous to use large or small server learning rates. Finally, there are some experiments concerning the use of random reshuffling and server learning rates.

**Broader Impact Concerns:**

None.

**Requested Changes:**

The primary change that the authors would need to showcase to secure my recommendation would be to showcase some setting in which random reshuffling is actually beneficial for the overall convergence rate. As discussed above, in the rates in Section 6, you basically always want to set $\gamma = 0$, in which case random reshuffling does not matter.

Second, I believe that the paper would need to have its empirical validation better reflect the core of what the paper is about. If the experiments are doing federated training, then clarifying the characteristics of that training would be useful. If not, then I believe it is necessary for this paper to have their experiments occur in actual federated contexts, given the focus of the paper.

Ideally, the experiments could also reflect the actual algorithm analyzed in this work. For example, Theorem 4 shows an interplay between large/small server learning rates and the sampling proportion of clients. Showcasing this in an empirical setting would be much more convincing than the set of experiments currently in the work. Highlighting interactions with heterogeneity (both inter-client and intra-client) as well as interactions between client/server learning rate and the efficacy of random reshuffling could also be interesting.

Last, I believe that this paper needs somewhat improved scholarship around the area. I would personally advocate for Table 1 to be removed entirely. Instead, I believe the authors should convey the same information by discussing related work in a positive manner (eg. Paper X first analyzed Y. We build on their analysis by incorporating Z, but use a different technique W). Similarly, I think that this paper cannot really claim that Nastya is a new novel algorithm. By simply renaming it to something more descriptive (rather than something aimed at associating it with this specific paper), such as "FedAvg with Random Reshuffling" would be beneficial.

### Minor comments

*  I really liked how the authors carried through the usage of color on the client and server learning rates throughout their exposition. This made it a lot easier to understand and compare things, though I wonder if there's a way to carry this over to people who have trouble distinguishing reds and greens.
*  The complexity improvements at the end of Section 5 are relatively opaque. They involve certain choices of parameters in the subsequent bounds in Section 6, and in a few cases I was not able to immediately derive them, especially as the choice of parameters needs to be dependent on the error threshold. Formalizing these complexity improvements at the end of section 6, with explicit hyperparameter choices would be greatly appreciated. Similarly, adding references for the quoted bounds of standard random reshuffling would be beneficial.
*  A paper on federated learning and shuffling was published in TMLR this month. I have not read the paper fully, so I cannot comment, but it seems like there are a number of parallels between this work and that. I do not know what TMLR's policy on things like concurrent submissions are, but trying to disambiguate them could be useful for a reader.

**Strengths And Weaknesses:**

I quite like the general thrust of the paper. In particular, the authors attempt to identify techniques that benefit federated optimization in practice, and analyze/explain them theoretically. The writing is also quite clear. Many federated learning papers are awash in opaque assumptions around heterogeneity, and reducing them to their core essence for interpretability is a nice feature. Moreover, as best as I can tell, the paper is sound, and the proofs are relatively easy to follow. The optimization bounds are relatively easy to interpret (though this comes potentially at a cost, see discussion below).

I also like the general idea of the work in trying to understand how different factors interact. There are a wide variety of techniques used in federated optimization that are inter-dependent. For example, client and server learning rates have their own interaction, but also interact with the participation rate and heterogeneity terms. Holistic understandings are certainly worthy goals of a paper in this area.

Unfortunately, there are a number of significant areas where I feel that this paper falls short of the mark. I discuss a few of these below.

## Utility of convergence results

The first issue I have with the paper comes from whether or not the convergence bounds are useful. Notably, the theory in Section 6 requires the client learning rate $\gamma$ to satisfy $\gamma \leq n\eta$ where $\eta$ is the server learning rate and $n$ is the number of examples held by each client. In settings where $n$ is at least moderate, this effectively means that $\gamma$ is close to 0 (and in fact, this is how the authors explain their theory in a cursory manner in Section 5). Looking at the convergence rates, it is clear that they are strictly minimized by setting $\gamma = 0$.

However, once $\gamma = 0$, FedAvg reduces to distributed SGD. What's more, the random reshuffling doesn't matter when doing distributed SGD, as we are simply summing gradients of examples taken at the same point. At this point, the convergence rates become well-known convergence rates for distributed SGD. All in all, section 6 does not give one a compelling reason to care about client versus server learning rates or random reshuffling. Taken at face value, we should simply avoid such concepts altogether and do distributed SGD.

The theory in section 7 is slightly more interesting. That is, the notion that the server learning rate should be set small if (for some reason) the client learning rate is large or the participation rate is very small. Again though, the theory does not elucidate why one would want to set the client learning rate to be non-zero in this setting.

## Experimental evaluation

Before I get into this, I will say that I understand that theoretical papers need not have robust, or even any experiments to be worthy of acceptance. That being said, this paper includes experiments, allegedly to validate some of the theory in more controlled settings, but personally I feel that these experiments don't contribute to what the paper is attempting to show.

First and foremost, it is unclear if the experiments are using federated learning at all. The authors mention 2 non-federated datasets. Thus, if the experiments are somehow splitting those into clients, then no details are included as to how that partitioning occurs. Similarly, there are no details on how many clients there are, what the partial participation rate is, or how heterogeneous the clients are. Note that all three of these terms are crucial to all of the convergence bounds in the paper. However, the discussion of the experiments (eg. referencing a number of epochs) makes me inclined to think these are centralized experiments, in which case it is not clear how they're even doing something like FedAvg.

Other details also seem irrelevant to the theory. For example, on the LibSVM dataset, they use an adaptive estimation of smoothness which is not centered in any of the theory. On the CIFAR-10 experiments, they use learning rate decay and Adam, neither of which are relevant to theory in question. While these techniques generally concern the reduction of learning rates (which is tangentially related to the question of small versus large server learning rates) it is difficult to create a coherent through-line between the theory and experiments here.

## Audience interest and paper contributions

Some of this work may be of interest to TMLR audience, especially the specific convergence rates derived. However, I believe that much of how the work (especially Sections 4 and 4) is structured may reduce general interest in the paper due to how the authors position it with respect to other works in the literature. In short, instead of positioning their research as generally something that builds upon years of optimization research in FL, they seem to suggest that this work is more complete than other works.

In particular, I am concerned with Table 1. Table 1 contains a number of influential works on distributed and federated optimization, and tries to categorize them according to what federated optimization characteristics (eg. partial participation) they cover. This is reductive at best, and potentially disingenuous. All of these works are multi-faceted efforts that contain a number of crucial observations that have moved the field forward. This paper has benefitted (even indirectly) from them. Moreover, these works were not necessarily about the same topics in this paper. I find this highlighting of such works only to say "they don't analyze this aspect" with a red X very strange, and something that would reduce interest among readers. Rather than building up a coherent picture of the history of this area, the related work section seems to only say in which ways this analysis is superior to prior works.

Similarly, I believe that the entire framing of the Nastya algorithm reduces reader interest in a similar fashion. As best as I can tell, Nastya is not a new algorithm. It is one of the multiple framings of FedAvg in terms of client and server learning rates (see Karimireddy et al., 2020 or Reddi et al., 2021), combined with random reshuffling. In particular, the FedOpt algorithm in Reddi et al., 2021 is strictly more general, and the empirical evaluation in that work uses random reshuffling at clients in their implementation of FedOpt. I frankly don't understand what the benefit is in trying to claim that one of the core contributions of this work is a "New Algorithm" (page 3). I think this simply detracts from the core theoretical optimization bounds in the work.

---

> ### Author Response · Authors · 2022-09-28
> **Thanks for the eloquently and thoughtfully written review**
>
> Dear reviewer,
>
> Thanks for such a well explained and reasoned review. It was a pleasure to read. In fact, I do not know when was the last time I've a read review like this in a ML conference. While I personally do not agree with all criticism (and I did not check with my coauthors yet), I understand all points made, and almost everything is quite spot on. I also very much appreciate the effort! We'll respond as soon as we can, but right now we need to put some final touches on a new paper with a deadline that is upon us.
>
> Kind regards,
>
> authors

---

### Review · Reviewer_eu8P · 2022-10-11

**Summary Of Contributions:**

The submission studies federated learning (FL) with realistic assumptions (e.g. partial participation) and analyzes the role/benefits of two techniques in this context: random reshuffling and server stepsizes. Specifically, it analyzes a variation of FedAvg called Nastya, which incorporates server stepsizes and local update directions.  The paper argues that both small and large server stepsizes can sometimes help. It also includes convergence analysis of Nastya with (strongly) convex and non-convex loss functions. Initial experimental results are also reported.

**Broader Impact Concerns:**

It might be worth discussing the privacy implications of federated learning. For example, local storage is insufficient to prevent client data from being leaked. The literature on *differentially private* federated learning would be relevant to point to.

**Requested Changes:**

The items above would need to be thoroughly clarified and the paper substantially revised in order to gain my endorsement for publication. I think a fundamental re-writing of the paper may be needed to hone the focus on one or two clear questions and provide a more clearly, carefully articulated, interesting answers to those questions. I suggest the authors put more thought into what their contributions are/how they are situated in comparison to works like SCAFFOLD and Woodworth et al., and around how to communicate these ideas clearly.

**Strengths And Weaknesses:**

**Strengths:** The idea of rigorously exploring the interaction between client and server stepsizes, and how to optimally tune them in a FedAvg-like setting seems like an interesting one. If this paper could have dived in deep on this and clearly provided the reader with an understanding of the delicate dance between client and server stepsizes, their dependence on heterogeneity, and if/how/when/why they can lead to improvements over standard FedAvg, then there would be potential for a strong paper. Similarly, the idea of carefully understanding random reshuffling seems interesting. Further, if there is some interesting connection or relationship between server-side optimization and random reshuffling, or clear practical guidance on how to optimally manage both, then that would make for a nice paper.

Unfortunately, these ideas were only superficially discussed and I did not end up with a clear or deep understanding of any of the above questions. In general, the ideas in the paper are scattered and the central message(s) is not clear or well-supported.  The paper feels like it is floating around in different topics like random reshuffling and server-side optimization without clearly stating what we learned about either of these things from the analysis or experiments.  In more detail, the following items were unclear from my reading:

**Weaknesses:**

*Abstract:*

The abstract uses terminologies (e.g. random reshuffling, local passes, server-side optimization) that is not standard/clear without defining them.

The abstract makes claims that are not clearly supported or discussed in the paper. The big claim that stood out in the abstract was "In particular, in non-convex regime we get an enhancement of the  rate of convergence from  $O(\epsilon^{-3})$ to $O(\epsilon^{-2})$." First of all, the meaning of this claim is not exactly clear. FedAvg does not have $\epsilon^{-3}$ convergence rate according to your Table 2. But more importantly, I did not see this claim discussed or justified in the main body of the paper. Making major claims in the abstract without careful explanation/justification in the main body is a serious deficiency.

The sentence "first time that local steps...help to overcome the communication bottleneck" is vague and its meaning is unclear.

*Introduction:*

I don't think subsection 1.1 is needed: you can flow from the first paragraph (about data) to the second (about federated learning) without a break.

The term "clients" is used before it is defined.

The FL examples and motivation for the paper all seem to be related to cross-device FL, not cross-silo FL. This should be stated explicitly.

*Ingredients of successful FL methods*

It is strange that "partial participation" is classified as an "ingredient" for success. I usually think of this as a *challenge* associated with FL that must be overcome: in an ideal world, if we could query all clients in every round, then we would, right? The authors claim that the "server chooses a cohort of clients" to participate  in each round. This is not accurate in general since oftentimes the network factors determine which clients are able to participate in a given round (e.g. due to internet connectivity for cross-device FL). The authors claim in passing that partial participation can be "useful" due to "diminishing returns": this claim is surprising and should be clearly justified and explained if it is to be included. They also claim that partial participation is a "necessity" in cross-device FL and that most users only participate once: this claim should also be supported with an explanation (e.g. is it necessary due to server compute limitations? why do most users only participate once?) and reference.

The first two sentences of the *data shuffling* paragraph do not seem to be related: what does batch size have to do with replacement (vs. no replacement) sampling? Also, the terminology "SGD" is used to refer to *with-replacement SGD*: this should be clarified, as SGD does not always need to implemented using with-replacement sampling. Further, it is suggested that without-replacement sampling leads to biased updates: this is a strange claim, as we often think of client $i$'s local data as being i.i.d. Where would the bias come from?

Server stepsizes: The meaning of the "it is better to aggregate model updates... routine" sentence is unclear.

*Summary of Contributions*

"We know virtually nothing about server stepsizes": This is a very strong and seemingly inaccurate claim. For example, the SCAFFOLD paper studied the role of server stepsizes in a FedAvg-type setting, did they not? Also, any paper (e.g. Woodworth et al, 2020) that analyzes MB-SGD would reveal some insights about server stepsize, since it is a key algorithmic parameter.

It is claimed that the key focus of the paper is on server stepsizes. I like this sentence because it grounds the reader and leaves him/her knowing what to expect. However, this focus is lost in the rest of the paper. Instead, random reshuffling is discussed/analyzed (e.g. Section 5) before we even see much about server stepsizes at all. Further, the discussion that there is around server stepsizes is unclear and does not seem rigorous. I discuss this weakness more below.

Table 2 is very unclear: i) Do the entries represent the number of communication rounds needed to get an $\epsilon$-accurate solution? Also, what does an $\epsilon$-accurate solution even mean (e.g. for nonconvex it would presumably mean something much different than for strongly convex). These important points are never stated. ii) there are problem parameters (e.g. L, $\kappa$, $\sigma_{*}$) used that are never defined until much later in the paper; iii) Local SGD is listed twice but other FL algorithms (e.g. MB-SGD, or AC-SA--which was shown by Woodworth et al to be *optimal* in many parameter regimes) are not included. Since Woodworth et al. provides algorithm-specific *upper and lower bounds* for Local SGD, it is not necessary to include two references/rows for Local SGD. iv)  Convex losses are not included in the table, which is strange.

*Complexity analysis*: What are we to learn from this? Does your method ever improve over Local SGD? Also, how does your complexity compare vs. MB-SGD and AC-SA and SCAFFOLD? From my own interpretation  of Table 2, it seems that Nastya is inferior to existing algorithms in the regimes $M, n \gg 1$, and I imagine it would be also be inferior to AC-SA in most regimes since AC-SA is provably optimal in many regimes.

There are ??'s in the references to Table numbers

*Small client stepsizes....drift reduction*: "eliminates the second of the three terms" sentence is not clear.

Theorem 5 is referenced in passing but the Theorem is in the appendix and not clearly discussed anywhere else in the paper.

*Experimental validation*: Figure 1 seems out of place in Section 2.  Too much detail for an overview of contributions. Instead, please explain at a high level what you found, then provide detailed illustration in the appropriate section of the main body.

*Lemma 1*: "population variance" is unclear/inaccurate: this is finite sample case.

*Definition 4*: "variance" is a strange name to use; the defined quantity does not appear to be the variance of any random variable.

*Nastya Algorithm*:

The gradient estimator in line 11 of Nastya is interesting, seems like a promising way to reduce variaance, especially in the presence of  client heterogeneity. I would like to have seen more discussion around this. For example, (how) does it differ from Scaffold?  (How) does this have theoretical benefits over standard FedAvg or MB-SGD?

*Section 5* especially "complexity improvements" (where you talk about special cases of your theory, which has yet to be formally presented) does not seem like it belongs before Section 6: Theory.

Where are the proofs of the results in "complexity improvements"? The argument given in Section 5 was not rigorous.

*Section 6*:

Theorems 1-3: the performance of the algorithm appears to be optimized by taking local stepsize = gamma = 0. This is very strange: if local stepsize goes to 0, then no progress is made by the algorithm.

Discussion about implications of Theorems 1-3 is severely lacking. Why are they interesting? What are the important takeaways? How do these results fit in with what prior works have shown?

There is a lot of discussion about random reshuffling, but where we see the benefits of random shuffling in Theorems 1-3?

What's the difference between Thm 1 and Thm 4? Both seem to give (different) bounds on the same quantity for the same algorithm and the same function class. Discussion is lacking.

"However, in some cases, our analysis shows that using small server stepsize and large client stepsizes can be beneficial and it means that we gain from using local steps" In what cases? Why?  There is some discussion around this at the end of the section, which is actually one of the stronger parts of the paper.

*Experiments*: they don't seem all that relevant to the rest of the paper. For example, "adaptive outer stepsizes are helpful when the base stepsize is not chosen well, which is in line with our theory." Is it relally in line with your theory? I didn't see any discussion earlier about what happens when user stepsizes are improperly chosen.

---

### Review · Reviewer_JW99 · 2022-10-28

**Summary Of Contributions:**

This paper provides a new theoretical analysis of the federated averaging algorithm, which combines partial client participation, local updates, random shuffling, and more importantly, server step size. The authors claimed that they propose a new algorithm (which I think is not true), get better complexity results and provide novel insights on the effects of server step size. Some toy experiments were used to validate the theoretical findings.

**Broader Impact Concerns:**

No concerns.

**Requested Changes:**

Please see the comments in the last section.

**Strengths And Weaknesses:**

## *Strengths*

- The theoretical analysis covers multiple different settings and use minimal assumptions on data heterogeneity.

## *Weaknesses*
- The motivation of this paper is a bit unclear. Basically, partial participation, local training, random shuffling, and server step size all have been studied before in literature either in isolation or in combination. I agree that it is nice to have a unified theory to cover all these components. But if this is a very easy thing to do, then it is not worth publishing a paper. It is possible that we just need to trivially combine different analyses together. The authors failed to show why this is a difficult problem. Like the authors mentioned, there are also other very useful tricks in federated learning, like server momentum. If the authors want to give a comprehensive study, why not also include server momentum?  I kind of feel the analysis is unnecessarily complex. If you just want to show the benefit of server step size, then it would be better to first have a very simple theorem under the simplest setting and then combine it with other components as an extension.
- Following the above point, I particularly did not get why random shuffling is needed in this paper, especially given the fact that an important lemma (lemma 1) is directly borrowed from a previous paper. If you want to show that the benefit of server step size can only be seen when used together with random shuffling, then the current analysis is not enough. You need to compare the analyses in both with and without random shuffling settings and show different effects of the server step sizes.
- I think in many places the authors oversell their contribution. This can make the readers very uncomfortable. For example, (1) they said, "we design a new algorithm, for which we coin the name Nastya". However, this algorithm is not new at all. It is just the practical FedAvg algorithm implemented in any FL package. (2) in the abstract, "this is the first time that local steps provably help to..". However, in previous paper like (Woodworth et al., 2020 & 2021) have already show this. Also, I didn't find any discussions in the paper to support this sentence. (3) "with a couple exceptions only (Karimireddy et al., 2020; Woodworth et al., 2020), there are no prior theoretical works analyzing the effect of server stepsizes in federated learning." I think this is also not true. Incorporating server step sizes into convergence analysis has already become a standard technique in literature, see [1] [2], also [3] studies it through a different perspective. It is well known that the optimal server step size is not 1. Given these prior works, it is unclear why it is still important to study the effect of server step sizes.
- The paper is not sufficiently polished.
    - The authors mentioned that they will discuss related works in the appendix. Unfortunately, the appendix does not discuss any related work.
    - It is unclear how the authors derive the convergence rate in table 2. Even in the appendix, there is no proofs showing how we get the rate from theorem 1, 2, 3. Similarly, the rate O(e^-3) O(e^-2) do not appear anywhere in the main paper. It is unclear how the authors derive them.
    - There are many question marks remaining in the paper.
- I did not get why the experiments on CIFAR-10 can support your theory. "Adaptive step sizes help when the base step size is not chosen well". This is common sense because adaptive optimizers are designed to reduce the hyper-parameter tuning burden. Also, on CIFAR-10 dataset, it is known that Adam is not better than SGD.
- Theorem 1, 2, 3 are a bit confusing. It seems that gamma = 0 can minimize the optimization error upper bounds. That means, there is no need to have any local updates. Performing batch GD is optimal. This is in conflict with practice and is in conflict with the sentence in abstract "local steps provably helps".

[1] Yang, Haibo, Minghong Fang, and Jia Liu. "Achieving linear speedup with partial worker participation in non-IID federated learning." arXiv preprint arXiv:2101.11203 (2021).
[2] Jhunjhunwala, Divyansh, et al. "FedVARP: Tackling the Variance Due to Partial Client Participation in Federated Learning." The 38th Conference on Uncertainty in Artificial Intelligence. 2022.
[3] Charles, Zachary, and Jakub Konečný. "Convergence and accuracy trade-offs in federated learning and meta-learning." International Conference on Artificial Intelligence and Statistics. PMLR, 2021.

---

### Decision · Action_Editors · 2022-12-11

**Recommendation:** Reject

**Comment:**

This paper has received three reviews of outstanding quality: all reviewers engaged with the content of the paper very deeply, verifying the technical results, assessing the quality of writing, and giving insightful comments on the related work. All reviewers took issue with the positioning of the results in the paper, pointing out that the authors may have mischaracterized the novelty of the results. Another common concern was that the presentation of the results is not particularly clear, and the paper would have benefitted from having a cleaner narrative that focuses on the parts of the analysis that are indeed novel (e.g., the effects of reshuffling). Some oddities around the theoretical results were also pointed out (e.g., that setting $\gamma = 0$ appears to optimize the rates, which contradicts both common wisdom and some key claims made in the paper).

The authors have only responded to one of the reviews, and even that without addressing the criticism of the reviewer in question. Based on the reviews and my own reading, the paper does seem to have many strong points, and it should be possible to turn it into a great paper once the narrative is straightened. Since the reviewers (and myself) are all in agreement about the shortcomings of the paper, and the issues they have pointed out are much more profound than what can be realistically addressed in a single revision, the best course of action at this point is to reject the paper. I encourage the authors to take the reviews seriously into account when preparing the next version of the paper.

**Audience:**

The research community working on federated learning would certainly be interested in the results presented in the paper.

**Claims And Evidence:**

The formal claims are all well-supported by the proofs, which appear to be all technically correct and well-written. However, several reviewers have pointed out that the positioning of the results is potentially misleading: the algorithm is apparently not as novel as the authors claim (it's just a variation of the standard FedAvg algorithm), and most of the algorithmic components involved have been studied in previous work. Table 1 and the discussion around it may have mischaracterized parts of the relevant literature.